# Transient infrared nanoscopy resolves the millisecond photoswitching dynamics of single lipid vesicles in water

T. Gölz [1], E. Baù[1], J. Zhang[2], K. Kaltenecker[1,3], D. Trauner [4], S. A. Maier [5,6], F. Keilmann [1] ✉, T. Lohmüller [2] ✉ & A. Tittl [1] ✉

Understanding the biophysical and biochemical properties of molecular nanocarriers under physiological conditions with minimal interference is critical for advancing photopharmacology, drug delivery, nanotheranostics and synthetic biology. However, analytical methods often struggle to combine precise chemical imaging and dynamic measurements without perturbative labeling. This challenge is exemplified by azobenzene-based photoswitchable lipids, which are intriguing reagents for controlling nanocarrier properties on fast timescales, enabling precise light-induced drug release. Here, we leverage the chemical recognition and high spatio-temporal resolution of scattering-type scanning near-field optical microscopy (s-SNOM) to demonstrate a non-destructive, label-free technique for mid-infrared imaging and spectroscopy of individual photoswitchable liposomes. Our transient nanoscopy approach enables imaging below the diffraction limit and tracks dynamics with sampling times as fast as 30 ms. We resolve photoinduced changes in shape and MIR spectral signature of individual vesicles and discover abrupt and delayed photoisomerization dynamics. Our findings highlight the method's potential for studying complex dynamics of unlabeled nanoscale soft matter.

Lipid-based nanocarriers, such as liposomes and lipid nanoparticles (LNPs), have emerged as a leading platform technology in nanomedicine[1,2]. Their key advantage lies in the ability to encapsulate hydrophobic drugs or molecular nanoagents for targeted delivery. Unilamellar vesicles, the most basic form of nanocarriers, have already found their way into clinical applications[3]. In addition, LNPs represent arguably the most advanced nanocarrier technology and have played a crucial role for the successful in vivo administration of mRNA-based vaccines[4].

Enhancing the performance of liposomal nanocarriers, including both liposomes and LNPs, depends on optimizing strategies to control site-specific release mechanisms and upon an external trigger[5]. Light is

ideally suited for this task due to its contactless nature and ease of focus. Photoswitchable molecules integrated into or forming part of the lipid membrane can facilitate release. Recently, Chander et al. made significant progress towards this goal by using the azobenzene-based photoswitchablephosphatidylcholines azo-PC[6], and red-azo-PC[7], in lipid nanoparticle formulations, enabling controlled drug release upon irradiation at specific wavelengths[8]. Their work demonstrates the feasibility of integrating photolipids into clinically approved lipid formulations, showing large promise for future development.

Imaging and spectroscopy techniques in the mid-infrared (MIR) spectral range are an ideal for investigating the chemical composition

[1]Department of Physics, Chair in Hybrid Nanosystems, Nano-Institute Munich, Ludwig-Maximilians-Universität München, Munich, Germany. [2]Department of Physics, Chair for Photonics and Optoelectronics, Nano-Institute Munich, Ludwig-Maximilians-Universität München, Munich, Germany. [3]Attocube Systems AG, Haar, Germany. [4]Department of Chemistry, University of Pennsylvania, Philadelphia, PA, USA. [5]School of Physics and Astronomy, Monash University, Clayton, VIC, Australia. [6]Department of Physics, Imperial College London, London, UK. ✉e-mail: fritz.keilmann@lmu.de; t.lohmueller@lmu.de; andreas.tittl@physik.uni-muenchen.de

                                    

of different organic and inorganic samples[9] due to the wavelength-specific absorption of infrared light by the chemical material's bonds, often referred to as the spectroscopic fingerprint. In the case of photoswitchable lipid membranes, MIR spectroscopy is particularly useful for analyzing the isomerization of the membrane-embedded photolipids in a label-free and non-destructive manner without interfering with the switching process itself[10,11]. However, conducting MIR imaging and spectroscopy on a single lipid vesicle requires a methodology that simultaneously provides sufficient nanoscale spatial resolution and high temporal resolution to resolve the photoisomerization dynamics. Scattering-type scanning near-field microscopy (s-SNOM)[12,13] is ideally suited for this task. In s-SNOM, a laser beam is focused onto the apex of a sharp metallic AFM tip creating highly confined evanescent fields around the apex that yield spatial resolutions down to 20 nm. The method has been highly successful for studying single biological macromolecules[14–16] and lipid monolayers[17] in air, and has already been extended to observing living biological entities in their native aqueous environment[18].

A critical gap in optimizing photolipid-nanocarriers has been the lack of effective tools for studying membrane photoisomerization at the single liposome level. Previous studies have shown that azobenzenes quench fluorescence and dye molecules further interfere with the isomerization process[19], which renders fluorescence-based methods less favorable. While atomic force microscopy (AFM)[20], interferometric scattering microscopy[21] and transmission electron microscopy (TEM)[22] allow for investigating liposome shape and size with sufficient resolution, they do not provide any chemical insights into the isomerization process. Nanophotonic based sensing approaches show great promise for spectroscopically tracking complex dynamics, but lack the simultaneously flexibility to image the system[23–26].

Here, we demonstrate the use of in-situ nanoscopy to image and spectroscopically analyze individual photoswitchable lipid vesicles with apparent sizes down to 176 nm in an aqueous environment. In contrast to previous investigations on naturally progressing biological systems[18], we present an in-situ s-SNOM study on actively induced dynamic processes by reversibly changing the morphology of a vesicle through repeated ultraviolet/blue light illumination and tracking its spectral response at 30 ms sampling time. Our method demonstrates not only the possibility to detect and distinguish two photoisomeric states of the lipid molecules on the single lipid vesicle level based on subtle changes in their near-field MIR spectra, but also monitor the photoinduced transformations of the lipids in their aqueous environment in real time.

## Results

### In-situ near-field MIR imaging and spectroscopy of a photoswitchable lipid vesicle

In our experiments, the MIR near field of an irradiated metallic s-SNOM tip probes a water-suspended lipid vesicle through a 10 nm thick SiN membrane, which protects it from the atmosphere above and where the vesicle remains adsorbed by van der Waals forces for extended periods of time (Fig. 1a). The advantage of using the SiN membrane is that it prevents sample evaporation while also protecting the tip from contamination from the solution (for experimental details, see Methods). Furthermore, the SiN membrane-based in-situ technique enables mechanically stable s-SNOM measurements for hours, without the need to realign optics even when changing samples[18].

The MIR laser beam is tightly focused by a paraboloidal mirror onto the tip to generate a highly concentrated near field under the apex (Fig. 1a, red area). The optical near field penetrates through the SiN layer and probes more than 100 nm into the underlying liquid compartment[18]. Back-scattered coherent MIR light is detected in a Michelson interferometer setup, which allows for the extraction of both the MIR amplitude $s_2$ and phase $\varphi_2$ (Fig. 1b and Methods). The

choice of (i) a tunable monochromatic MIR laser or (ii) a wideband coherent MIR supercontinuum source allows for the assessment of either (i) the sample's response at a selected molecular vibration with high speed or (ii) the sample's full molecular fingerprint in a matter of seconds to minutes. The latter modality is known as nano-FTIR[12,27]. An additional mode of MIR operation is the white light recording of the detector signal, where all spectral components of the wideband continuum contribute, forming a spectrally averaged infrared signal. The reversible photoisomerization is induced by light from either of two LEDs emitting at 365 nm and 465 nm, respectively. Both LEDs are aligned to illuminate the entire SiN membrane area homogeneously without generating enhanced near fields (Fig. 1a, blue area, Fig. 1b).

The vesicles under investigation are composed of a 50:50 % mixture of 1,2-dioleoyl-sn-glycero-3-phosphocholine (DOPC) and azo-PC (see Fig. 1c and Methods for details on sample preparation)[6]. In general, azo-PC is a membrane forming lipid and both stable liposomes and supported bilayer membranes can be formed entirely from pure azo-PC. However, the 50:50% lipid mixture was chosen, since highest change in bending rigidity is expected based on previous micropipette aspiration measurements[28]. That way, both spectral changes due to photolipid isomerization, as well as mechanical vesicle deformations are expected, which aids in showcasing the full characterization potential of our approach. The switching wavelengths were chosen from the azo-PC UV-VIS spectra (Fig. 1d). Illumination of the lipid vesicles with a wavelength of 365 nm triggers the isomerization of the azobenzene moiety from *trans* to *cis* (resulting in a change of the bulky phenyl rings from being located opposite of each other to besides each other and drastically increasing the spatial footprint of the tail group of the lipid). Conversely, illumination at 465 nm switches the molecules to the thermodynamically more stable *trans*-state.

The photoswitching of azo-PC lipids has been analysed by bond-sensitive MIR spectroscopy[10,11], as we confirm for our samples using ATR-FTIR spectra of dried azo-PC films (Fig. S1). The carbonyl band at 1735 cm$^{-1}$ remains unaffected by the photoswitching[6,10], whereas clear differences in spectral intensity between the *cis/trans* isomers are evident from the resonances at 1602 cm$^{-1}$, 1496 cm$^{-1}$ and 1466 cm$^{-1}$, which can be assigned to the ring breathing mode of the *trans*-azo-group and CH$_2$-backbone modes found in the azo-PC lipids[11]. All studied resonances feature increased absorption in the *trans*-conformation[11]. In contrast, DOPC lipid ATR-FTIR spectra do not change in intensity with different illuminations and do not show a band at around 1600 cm$^{-1}$ (Fig. S1).

The in-situ s-SNOM technique enables MIR spectroscopy on individual lipid vesicles in an aqueous environment at length scales impossible to reach by standard far-field spectroscopy approaches. Since the most sensitive lipid resonance to the photoisomerization located at 1602 cm$^{-1}$ would be masked by a strong H$_2$O vibration at 1650 cm$^{-1}$, we suspended all vesicles in this work in D$_2$O, a common practice in FTIR spectroscopy of organic materials[18,29] (see Fig. S2 for spectra of vesicles in H$_2$O).

We first present a spectrally averaged white-light MIR near-field amplitude image (Fig. 1e) (second demodulation $s_2$, see Methods) to identify and target a SiN membrane-fixed vesicle in aqueous solution for further spectroscopic measurements.

We subsequently recorded nano-FTIR phase spectra $\varphi_2$ at the vesicle's center (Fig. 1e red cross and 1 f), where the highest scattered white-light signal is obtained and therefore a high signal-to-noise ratio (SNR) of the recorded spectra is expected (see Methods). In both photoisomerization states, these spectra show the known characteristic resonances of the lipid system, such as the carbonyl peak, the CH$_2$-backbone modes[10,11], and a resonance at 1606 cm$^{-1}$ (see Fig. S2 for a detailed assignment to the molecular bonds of DOPC and azo-PC). Notably, the resonance at 1606 cm$^{-1}$ decreases in intensity after illumination at 365 nm, indicative for *trans*-to-*cis* isomerization. In addition, the CH$_2$-backbone modes at 1466 and 1496 cm$^{-1}$ also decrease in

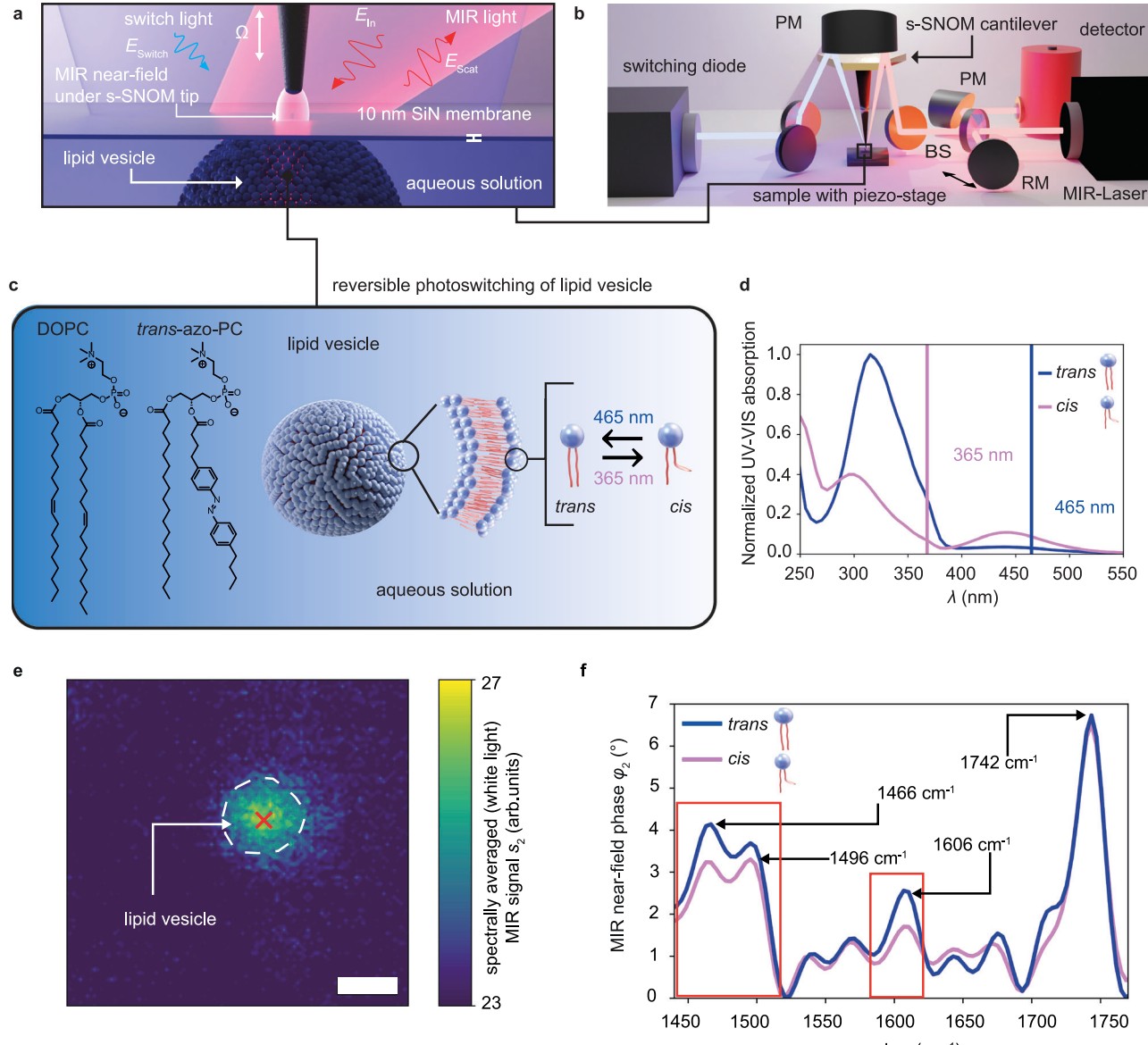

**Fig. 1 | In-situ s-SNOM infrared spectroscopy of photoactive lipid vesicles.**
**a** Operating principle of the membrane-based in-situ s-SNOM method. An s-SNOM tip with its associated near field scans in tapping mode at a tapping frequency Ω above a 10 nm thin SiN membrane, which separates the tip from lipid vesicles suspended in an aqueous medium. The tip and membrane are illuminated by two different light sources: an MIR laser beam ($E_{in}$) for near-field spectroscopy and imaging and a UV-VIS source ($E_{switch}$) to switch the lipid vesicles between their different photoisomeric states. **b** S-SNOM setup with the MIR beam and the UV light focused onto the s-SNOM tip and sample by a parabolic mirror (PM). The focused MIR beam creates an enhanced near field that interacts with the sample under-neath. The MIR light backscattered from the tip ($E_{scat}$) containing the spectroscopic

information of the liquid sample is collimated and interferes with a reference beam that is reflected by a movable reference mirror (RM). The resulting signal is recorded by a fast response infrared detector (s. Methods). **c** Molecular structure of the DOPC and *trans*-azo-PC lipids constituting the lipid vesicle in a ratio 50:50 and sketch of the light-induced conformational change. **d** UV-VIS spectrum of both lipid isomers with switching wavelengths labeled by the blue and violet vertical lines. **e** Spectrally averaged MIR amplitude image ($s_2$ white-light image) of a lipid vesicle in *trans*-state in $D_2O$, scale bar 500 nm. **f** MIR near-field phase spectra ($\varphi_2$) of a *trans*-(blue) and *cis*-state (violet) lipid vesicle (**e**) recorded at the position of the red cross. Red boxes highlight two lipid vibrational MIR bands that respond strongly to the switching light.

intensity. These differences are consistent with our measurements of ensemble averaged far-field ATR-FTIR spectra (Fig. S1) and with reports in literature[10,11]. Importantly, the carbonyl signal at 1742 cm⁻¹ in the nano-FTIR spectrum under 365 nm illumination does not change in intensity compared to 465 nm illumination, as is expected from the reference ATR-FTIR spectrum. This confirms that there was no change in laser power, optical tip alignment, or detector responsivity between the recording of the two spectra, which could lead to intensity mod-ulations. Therefore, the spectra demonstrate our capability to analyze the chemical composition and distinguish between photoisomers of a

nanoscale lipid vesicle by their associated nano-FTIR spectra. Note that the spectroscopic signal of a lipid vesicle in $H_2O$ outside the 1650 cm⁻¹ $H_2O$ band (Fig. S2) is of similar good quality and should therefore allow future studies in $H_2O$ suspension.

Based on the nano-FTIR spectra recorded on the lipid vesicle, we chose the intense carbonyl resonance to record resonance-specific MIR images. A larger area scan (15 μm x 15 μm) demonstrates the side-by-side coexistence of numerous vesicles of varying sizes, simulta-neously measured in both amplitude $s_2$ (Fig. 2a) and phase $\varphi_2$ (Fig. 2b). Individual vesicles can be clearly identified and localized, as

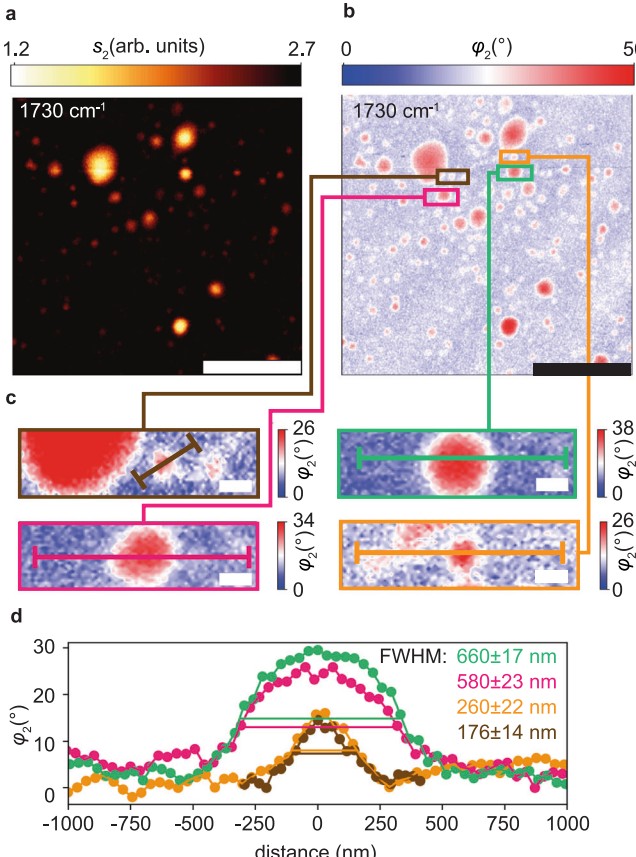

**Fig. 2 | Chemically specific MIR near-field imaging of nanoscale lipid vesicles.** S-SNOM optical amplitude ($s_2$) (**a**) and phase images ($\varphi_2$) (**b**) of several lipid vesicles suspended in $D_2O$ measured at 1730 $cm^{-1}$ at the carbonyl resonance (see Fig. 1f), scale bars 5 μm, acquisition duration 12 min. **c** Enlarged phase images ($\varphi_2$) of four differently sized lipid vesicles, scale bars 300 nm. **d** Extracted profiles along lines drawn in (**c**), with full widths at half maximum (FWHM) and error margins determined by Gaussian fits (not shown).

exemplified by close-up views of four differently sized vesicles, indicated as colored boxes (Fig. 2c). The vesicles typically exhibit near-uniform phase $\varphi_2 > 25°$ throughout their inside (shown in red), surrounded by a fringe of around 100 nm width (shown in white).

We extract quantitative phase profiles along the lines indicated in Fig. 2c and determine each vesicle's apparent full width at half maximum (FWHM, Fig. 2d). Despite only being 176 nm in width, the smallest analysed lipid vesicle is still well resolved (albeit with reduced phase signal), which is consistent with the spatial resolution of our setup on the order of 100 to 150 nm[18]. Importantly, this result demonstrates the capability of our method to characterize nanoscale objects even in water, at a length scale beyond the reach of standard fluorescence and phase contrast microscopies.

The uniform phase within and amongst differently sized vesicles (Fig. 2b) indicates that they undergo a deformation and flattening when adhering to the SiN membrane, as sketched in Fig. 1a. This interpretation of the measured uniform signal is in accordance with previous subsurface s-SNOM studies, which showed that the near-field sensing reaches to around 100 nm depth, but objects beyond 200 nm below the tip remain virtually invisible[30,31]. Therefore, it is safe to assume that the recorded signal originates solely from the adsorbed lipid membrane of 4 nm thickness[32].

To corroborate this flattening behavior, we compare phase profiles of the 660 nm FWHM vesicle and the largest vesicle (next to the brown box in Fig. 2b) with an analytical model for the phase signal of a

nondeformable polymeric sphere hanging from a single adhesion point on the SiN membrane. We find that the theoretical profiles have a different shape and are narrower than the experimental profiles (Fig. S3b, c), supporting the hypothesis of vesicle flattening. This observation demonstrates our method's ability to optically study details of nanoscale adhesion of soft matter systems in aqueous environments.

Independent further information about the vesicles can be gained by examining the simultaneously acquired *mechanical* images or vesicle footprints[18], which show a displacement of the SiN membrane upwards by about 1 nm for the green-boxed vesicle location and up to 2 nm for the largest vesicle (Fig. S3d, e). This observation clearly shows that even nanoscale soft-matter objects can be detected via distinct deformations of the SiN membrane, in agreement with previous studies[18,30], and should be applicable to characterize even more complex lipid systems.

### MIR near-field imaging resolves the reversible photoswitching dynamics of a single lipid vesicle

An approximately 500 nm wide vesicle was selected to map photoswitching-induced morphological changes by recording MIR images at 1603 $cm^{-1}$. (Fig. 3a, b). Each pair of images was acquired during intervals of approximately 2 min, with illumination periods of at least 1 min for inducing the photoswitching between image acquisitions. The time series commences with a round vesicle in the *trans*-state (Fig. 3a, b), followed by a blue/UV light illumination sequence to switch the photolipids multiple times between *trans* and *cis*. During the photoisomerization steps, reversible changes of the vesicle shape, size, and directionality of deformation were observed.

These size expansions of the vesicle when switching from *trans* to *cis* are in good agreement with previous reports, where it was shown that vesicles undergo transformations into less symmetric shapes following *trans*-to-*cis* isomerization[6], and that the area per photolipid increases by about 20%[33] to 25 %[34] for *cis* photolipids due to a higher packing density of *trans*-azo-PC in a lipid membrane. For the *cis* isomer, the conformational change of one lipid tail (see Fig. 1c) would increase the chain volume, with the lipids assuming a slightly inverted wedge shape. This change in packing density is reflected in the morphological change of the vesicles, which requires a rearrangement of the lipid molecules.

To confirm that we can monitor reversible photoswitching over long times and investigate a different lipid configuration in the form of a supported membrane patch adhering to the SiN membrane, we conducted a similar photoswitching time series (Fig. S4). The results again highlight reversible area changes due to photoswitching over four cycles within one hour, confirming the high stability and reproducibility of the s-SNOM measurements and the good reversibility of the switching process.

Furthermore, the time-series images (Fig. 3a, b), were used to extract both the vesicle's area $A$ and its circularity as a figure of merit for the asymmetry between both states as defined in Fig. 3c. Circularity is a commonly employed shape factor in image analysis[35,36] that numerically describes the comparability of the vesicle to a perfect circle. The vesicle resembles a perfect circle at a value of one and deviates from the ideal circle shape with a lower value. We delineate the vesicle's boundary by setting a threshold value of $s_2 = 7.8$ arb.units This value was chosen to encompass the area where the optical amplitude is above $\frac{1}{e}$ of its maximum value at the particle center after background subtraction. The changes between both photoisomerization states are significant and well reproduced, amounting to an increase of 10% in area and a decrease of 8% in circularity for the *trans*-to-*cis transition*.

One possible explanation for the change in circularity is that the 50% mixture of azo-PC and DOPC used in our samples displays lower bending rigidities (which quantifies the energy needed to change the membrane curvature[37]) in the *cis* compared to the *trans*-state, which

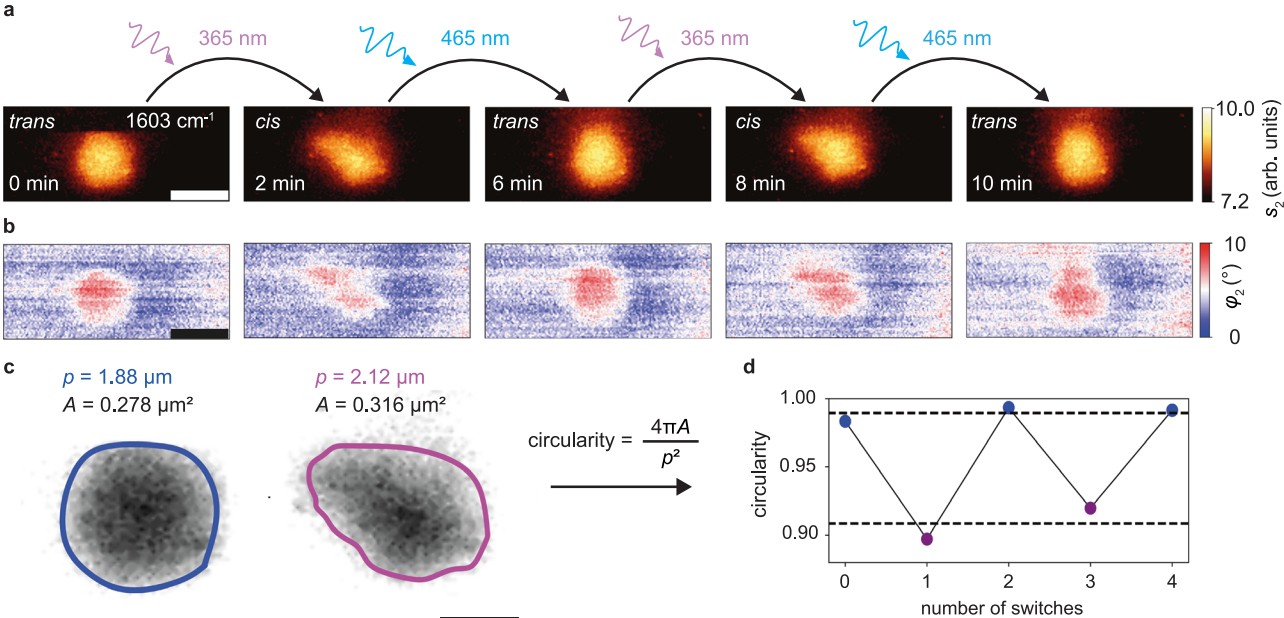

**Fig. 3 | Near-field imaging of the reversible photoswitching of a 500 nm diameter lipid vesicle. a** Monochromatic MIR amplitude ($s_2$) and **b** phase images ($\varphi_2$) at 1603 cm⁻¹ of a lipid vesicle in $D_2O$ being reversibly photoswitched between the *trans*-state and the *cis*-state, scale bars 500 nm. **c** Illustrated extraction of a circularity value for a lipid vesicle defined as $4\pi A/p^2$, with circumference $p$ and area $A$, where the boundary criterion was $s_2 = 7.8$ arb.units, scale bar 300 nm. **d** Reversible change in circularity between the *trans*- (blue) and *cis*-state (violet) of the lipid vesicle over two switching cycles.

aids vesicle deformation and explains the observed change in morphology[38]. Additionally, the photostationary state (PSS), defined as the *trans*-to-*cis*-ratio that is asymptotically reached by the photo-isomerization process, may not be quantitative, meaning that not all azo-PC lipids assume a 100% *trans*- or *cis*-conformation due to photoisomerization. The PSS is influenced by many experimental parameters, including the solvent, temperature, and the illumination conditions. For pure azo-PC vesicles in water, it was shown by small angle X-ray scattering that the fraction of *cis*-lipids in the blue adapted state (after 465 nm illumination) is still about 30%, while a considerable fraction of about 27% of the photolipids remain in a *trans*-state after UV illumination[32]. Furthermore, *trans*-lipids form H-aggregates, which may lead to demixing of *trans*-lipids into different domains on the vesicle membrane, contributing to the observed shape change[28]. Therefore, our method opens the door to studying such complex photoinduced deformation effects on a multitude of other nanoscale systems such as photoswitchable metal-organic frameworks[39] and nanoparticles[40].

### 30ms resolved MIR near-field signal tracking of the photoswitching dynamics of single lipid vesicles

Resolving the photoswitching dynamics of single lipid vesicles requires nanoscopy techniques capable of capturing fast structural and chemical changes. However, standard s-SNOM imaging is limited by slow mechanical scanning speeds, even for comparatively short per-pixel signal integration times $t_p$ resulting in long imaging durations of, for example, 50 s for a sub-µm lipid particle in water (Fig. 3a, b), where 150 by 100 pixels were acquired using $t_p = 3.3$ ms.

To overcome this inherent limitation of scanning probe imaging, we implemented a transient MIR nanoscopy method suitable for both aqueous and dry samples. This method continuously records the s-SNOM signals $s_2$ and $\varphi_2$ at a fixed location and set integration time. A specific wavelength is selected to obtain the maximum spectral response to photoswitching, and the tapping tip is placed on the center of a selected vesicle and remains stationary during the recording. This approach allows to resolve the switching dynamics at a sampling time down to $t_p = 30$ ms with SNR of 4 (Figs. S11, S12, S13).

A suitable vesicle is first identified by recording an MIR image at 1603 cm⁻¹ (Fig. 4a, d). We then confirm the vesicle's photoresponsivity by imaging the morphological distortion of the vesicle during photoisomerization. The tip is positioned on the vesicle's center (red and black crosses in Fig. 4a, d) to minimize signal distortion due to movement of the vesicle. The s-SNOM signals are continuously recorded at 1603 cm⁻¹ with $t_p = 500$ ms sampling time (Fig. 4b, e). The recording began after illuminating the sample for over 1 min at 465 nm, followed by switching once every full minute between 465 nm and 365 nm (blue and violet parts of the signal trace in Fig. 4b, e). Across the eight consecutive switching experiments, the signals consistently show an 8% decrease in amplitude and a 1.6° decrease in phase under 365 nm illumination, i.e., for *trans*-to-*cis* switching, consistent with ATR-FTIR and nano-FTIR spectra (Fig. 1f, Fig. S1). Comparable increases are observed for *cis*-to-*trans* switching under 465 nm illumination. The results demonstrate that the MIR near-field signals can indeed quantify consecutive switching events and thereby confirms their repeatability.

Notably, after an initial slow change, the MIR signals of each switching event in Fig. 4b, e exhibit abrupt accelerated changes lasting as little as 1 s in some cases. When fitting the MIR signals of the switching events in Fig. 4b, e with a sigmoidal response function (Figs. S5 and S6 and red curves in Fig. 4b, e), characteristic delay times $t_d$ ranging from 10.3 to 15.7 s and growth parameters τ ranging from 0.7 to 3.9 s could be extracted.

Both infrared signals originate from repeated changes in the local dielectric function of the material underneath the tip[41]. The dielectric function in the present case of organic soft matter is determined by intra- as well as intermolecular bonds[42] of the phospholipids. Quite generally, biochemical processes in liquid environment can be tracked quantitatively by our transient nanoscopy method, provided they change the dielectric function at a selected MIR frequency sufficiently strongly to overcome the instrument noise.

Note that the extracted photoswitching times are inherently convoluted with the sampling time of 0.5 s. Given that the observed growth parameters are approximately on the same order (between 0.7 s to 3.9 s) as the instrumental resolution, we expect the effect of convolution on the fitted values for τ to be non-negligible. This means

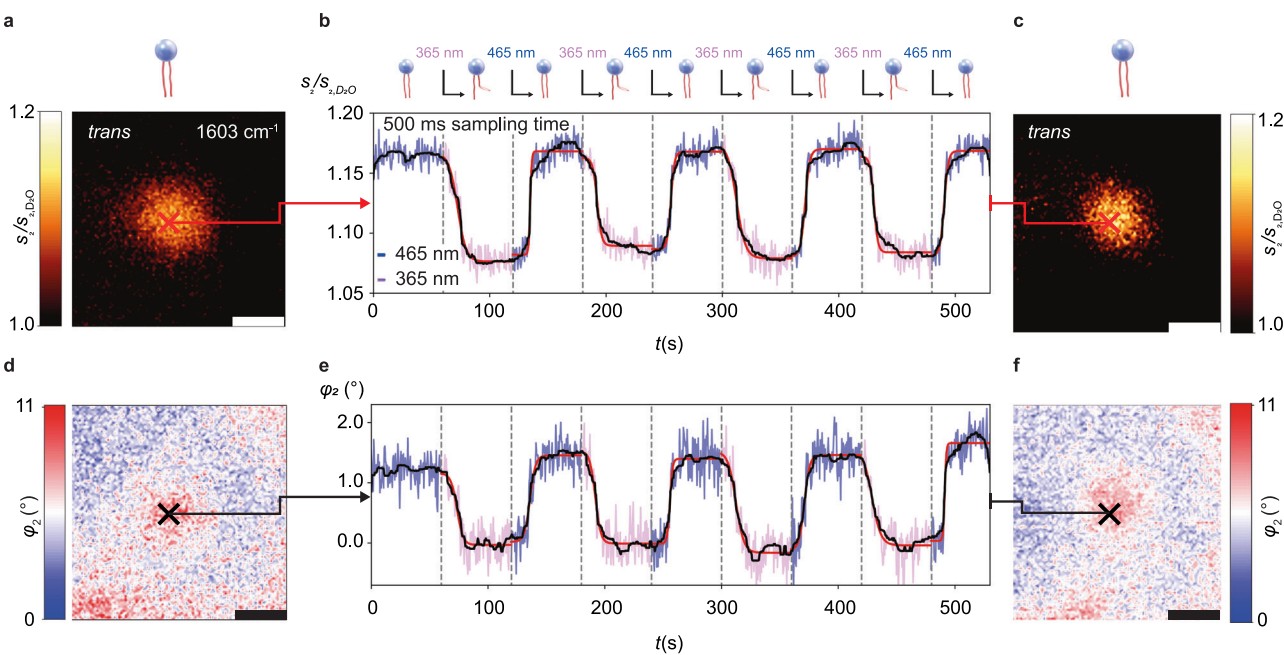

**Fig. 4 | Resolving the photoswitching dynamics of a single 500 nm lipid vesicle by MIR near-field signal traces. a** Optical near-field amplitude image normalized to the surrounding D₂O ($s_2 /s_{2, D2O}$) and **d** near-field phase image ($\varphi_2$), recorded at 1603 cm⁻¹ with 2 min acquisition time, directly before starting the MIR signal trace acquisition at the position of the cross, scale bars 500 nm. The recorded amplitude ($s_2 /s_{2,D2O}$) in (**b**) together with the phase ($\varphi_2$) in (**e**) reveal the photoswitching dynamics of the eight switching events at a sampling time of $t_p = 500$ ms. The symbols above (**b**) in combination with the dashed vertical lines indicate when the illumination wavelength was switched to the value written above. The blue and violet coloring of the signal traces mark the switching light's wavelength at each point. The black curves are moving averages of the measured data, and the red curves are sigmoidal fits of the switching process of the form $f(t) = \frac{L}{1+e^{\frac{(t-t_d)}{\tau}}} + C$ as displayed in detail in Figs. S5 and S6. **c** Optical amplitude and (**f**) phase images taken directly after acquiring the near-field traces show that the vesicle remained stable in both position and signal strength, even after eight consecutive switching transitions, scale bars 500 nm.

that the actual switching dynamics are likely faster than the fits would suggest, especially for lower values of τ that are closer to the instrumental response time.

This overall sigmoidal behavior with delayed, abrupt changes suggests that cooperative effects between the lipid molecules may play an important role during the membrane photoswitching process. The appearance of sigmoidal response functions has been previously demonstrated in related systems to describe cooperative effects, such as e.g., binding between giant unilamellar vesicle (GUV) membranes and polymer through domain formation[43], which has directly been linked to the presence of cooperative processes. Similarly, the dynamics of polymer-induced bursts of GUVs have also been found to behave sigmoidally due to the accumulation of polymer chains in the vicinity of the bilayer surface leading to a cooperative behavior that causes a sudden rupture in bilayer membranes[44].

Non-exponential and delayed-onset transitions have been observed for pure azo-PC vesicles by ensemble-averaged UV absorption measurements in liquid suspension[28,33]. During photoisomerization, abrupt changes of the lipid conformation along with a change of intermolecular interactions between *trans*- and *cis*-azobenzene embedded in a DOPC environment affect the isomerization dynamics[45]. In our experiments, the accelerated signal changes often occur between three or four data points separated by 0.5 s, i.e., within 1.5 or 2 s. An interesting perspective for future studies could be to reduce the illumination intensity, resulting in possible slower switching dynamics and thus allowing deeper insight into this acceleration phenomenon[33].

Cooperative effects promoting stronger lipid-lipid interactions of azo-PC in the *trans*-state have been reported in the literature. For example, the azobenzene groups of the lipid tails align side-by-side in a bilayer setting, which promotes the formation of H-aggregates[28,46].

Owing to their molecular conformation, *trans*-photolipids are also stacked more densely, which is reflected by a lower membrane diffusion coefficient of *trans*-azo-PC compared to its *cis*-form[38]. At the onset of UV illumination, the *trans*-*cis* ratio changes towards a *cis*-rich environment. Therefore, photoisomerization is delayed in the beginning, where a lipid conformation change is sterically hindered in the dense membrane assembly. In the extreme case of self-assembled azobenzene monolayers it has been shown that steric hindrance can even prevent the *trans*-to-*cis* isomerization altogether[47]. As the membrane shifts towards a *cis*-rich state, more space is available, which facilitates the overall switching of the azobenzene tails. In general, *cis*-to-*trans* isomerization is faster, as lipids are switched to their thermodynamically favorable state, although switching rates are strongly dependent on the illumination conditions and light intensity[33]. The extracted growth times τ from the sigmoidal fits of the time traces (Figs. S5, S6) indicate a higher switching speed in the *cis*-to-*trans* direction in comparison to the *trans*-to-*cis*-direction, in agreement with previous reports. Furthermore, a delay was also observed for the onset of *cis*-to-*trans* isomerization, which likely originates from membrane reorganization taking place as the photolipids reduce their lipid footprint[33], followed by a re-shaping of the entire liposome (Fig. 3). The abrupt change of the isomerization curve observed in both directions (Fig. 4b, e) has not been that distinct in absorption measurements of vesicle solutions. However, our findings are consistent with the switching dynamics observed by time-resolved monitoring of supported photolipid bilayer (SPB) isomerization by single particle plasmonic sensing[48]. For plasmonic sensing on SPBs and in the case of SNOM-measurements on a single vesicle, the illumination intensity is homogeneous over the entire sample. In comparison, for bulk measurements on ml volumes, performed in a cuvette, homogeneous illumination conditions are challenging to obtain. This supports our

understanding that nanoscopy with real-time information obtained from a localized membrane area under homogeneous illumination, provides further insight to otherwise hidden details of photo-isomerization dynamics of photolipids in a bilayer assembly.

This back-and-forth switching could be continuously monitored over an 8 min duration, encompassing four complete switching cycles. In addition, the time traces can be reproduced with good quality on the same sample (Fig. S7). After recording the time trace, we recorded another MIR image (Fig. 4c, f) to verify that the vesicle has not shifted in position relative to the tip, thus eliminating the possibility of signal fluctuations due to vesicle displacement during the switching process.

For comparison, we recorded a second transient signal trace at 1730 cm$^{-1}$ (Fig. S8), probing the carbonyl resonance that should remain unaffected by the photoswitching-induced molecular changes observed in Fig. S1. We measured only relatively small amplitude and phase signal changes, verifying that at 1603 cm$^{-1}$ we did probe the molecular switching. We attributed the recorded decrease in amplitude and phase from the *trans*- to the *cis*-state to a reduced lipid density in the *cis*-state[33] Moreover, we recorded a signal trace on D$_2$O next to the vesicle (Fig. S9) and on a clean silicon surface (Fig. S10). Both traces exhibited stable amplitude and phase signals, confirming that the measured switching signals are not due to mechanical or optical artifacts induced by the switching light.

To assess the temporal resolution limits of our in-situ tracking of dynamic processes, we repeated photoswitching experiments at varied integration times $t_p$ ranging from 500 to 30 ms on a second vesicle on a different day (Figs. S11, S12, S13). The signal-to-noise characteristics allowed to detect a photoswitching signature at a sampling time as short as $t_p = 30$ ms with a statistical significance limit of 3.72 $\sigma$ (see Fig. S11 and Figs. S12 and S13 for exact values). Notably, by increasing $t_p$ to 100 ms (Fig. S11 and S12) the statistical significance of the detection rises to 5.5 $\sigma$, showing the great potential for the technique to also be applied to dried samples such as, for example, the investigation of photoactive proteins at the single-molecule level[29]. Since these experiments do not need to be conducted in aqueous environment, even higher SNR than shown here could be expected.

## Discussion

By combining MIR near-field microscopy with ultrathin membranes and fast signal acquisition methods, we have demonstrated the label-free dynamic imaging and spectroscopic detection of actively triggered photoswitching processes in lipid nanovesicles as small as 176 nm in their native aqueous environment. This goes significantly beyond current state-of-the-art near-field microscopy studies on lipids, since spectroscopy, imaging, and transient signal tracing were conducted in liquid, enabling the efficient recording of dynamic processes. Specifically, our method allowed us to resolve reversible photoinduced shape changes in vesicles over multiple switching cycles. This capability enables future studies of shape effects in photorelease processes for a wide range of nanocarriers. Furthermore, leveraging the inherent spectral and therefore chemical sensitivity of MIR nanoscopy, even at the single lipid vesicle level, we could differentiate two main photoisomeric states of the photoswitchable vesicles based on nano-FTIR spectra. The chemical specificity of this method holds great potential for investigating other complex lipid systems on the nanoscale, such as the chemical composition of domains in lipid vesicles[28] or dynamics of lipid nanoparticles loaded with therapeutic compounds like mRNA. Here, one could consider first forming SPBs on the SiN membrane and then studying the fusion of liposomes with the lipid bilayer, mimicking a cellular environment and avoiding the problem of the vesicles adhering to the SiN membrane, which may influence isomerization dynamics.

Additional information about the geometry of the vesicles was obtained by comparing measured phase profiles of a single vesicle with an analytical model for a nondeformable sphere, suggesting a distinct flattening of the vesicles when adhering to the membrane. The vesicles were also identifiable in the correlative mechanical images, where we attribute the upward displacement of the membrane to van der Waals forces exerted by the vesicle. A potential solution to removing unwanted topographical artefacts caused by surface roughness is to use image reconstruction algorithms to correct optical images via the simultaneously measured topography[49], as well as improving fabrication methods. Additionally, more sophisticated approaches may be used in future studies to extract the permittivity and model the optical response of various biological materials adhering to the membrane through iteratively measuring and optimizing the liquid cell[50], which can serve as a tool to analytically calculate and predict the response of complex samples with an unknown dielectric function.

Importantly, we implemented a transient MIR nanoscopy technique to extract the non-linear switching dynamics of a single lipid vesicleat millisecond sampling time, achieved in spite of their rather weak backscattering.

One way to increase the scattering signals and with it the time resolution by as much as an order of magnitude could be the use of broader tips (albeit leading to reduced spatial resolution)[51]. With other, more strongly scattering objects, the time resolution might potentially improve to the low µs range.

This could be particularly interesting for electrically modulated solid-state materials[52], and would require higher phase modulation frequencies offered by photo-elastic or acousto-optic modulators[53,54]. Moreover, improvements in MIR laser stability and power, in addition to optimizing collection optics, could increase the SNR and lead to shorter acquisition times.

There are important performance differences between our membrane based liquid s-SNOM method and competing methods, which include in-liquid s-SNOM. In general, approaches that measure s-SNOM or AFM-IR directly in liquid can provide spatial resolutions only limited by the tip and recording of the full topography[55–59], whereas membrane-based method is limited to around 100 nm. However, these approaches all employ an ATR-based transmission illumination, which is very difficult to align. Moreover, our method enables more robust and longer duration experiments, because the tip is protected from contamination by the sample and the optical alignment is maintained when changing the sample, allowing more complex dynamic studies of, e.g., living cells[18]. Thinner membranes could be utilized to further increase the spatial resolution limit of our method, as long as they provide sufficient mechanical stability for water adhesion and AFM tapping. For example, liquid s-SNOM studies have been conducted with ultra-thin graphene capping layers[60–62] and metal oxide capping layers such as TiO$_2$ and Al$_2$O$_3$[63], which enable higher resolution imaging in liquid. Building on this idea, it has been demonstrated that a liquid sample containing viruses can be wrapped in a graphene sheet and investigated with s-SNOM enabling true nanoscale imaging and topography recording with a capping layer[64]. These capping layers have the additional benefit of having no phonon resonance in the MIR compared to SiN, which exhibits a strong resonance in the range of 800 to 1150 cm$^{-1}$ masking important molecular infrared resonances[18]. From a technological perspective, the wide and cheap commercial availability of the SiN membranes enables straightforward consecutive experiments on complex biological systems, as opposed to other capping layers, which need to be self-fabricated for every experiment.

In applications requiring super-resolution imaging, using finer tips in our s-SNOM measurements (ideally down to about 30 nm) in combination with thinner membranes could significantly improve both the optical and the mechanical spatial resolution. This would also reduce the near-field probing depth to 30 nm as the near-field probing depth is highly dependent on the tip[18]. The small probing depth would be ideally suited to image the adhering portion of a vesicle's membrane with improved contrast. Additionally, this

approach could be key to detecting inhomogeneities like lipid rafts and other nanostructures[51].

Note that in principle, our method is not limited to either relatively slow imaging (Fig. 3a, b) or fast one-pixel tracing (Fig. 4b, e); rather, it could combine both for a truly spatio-temporal assessment in real time. When extended to perform rapidly repeated, short line scans across a vesicle's edge[18], our method could shed light on whether abrupt spectroscopic changes inside the vesicle coincide with simultaneous morphological changes at its boundary in a correlative manner[18].

As a future prospect, the stable membrane-based in situ s-SNOM technique demonstrated in this work can be integrated with state-of-the-art microfluidics and environmental controls. This integration holds great promise for the nanoscale investigation of dynamic biochemical processes actively triggered by changes in temperature, pH, osmolarity, or the injection of chemical compounds. Additionally, our method is compatible with correlated measurements such as fluorescence and Raman imaging, as well as for THz nanoscopy, for which higher tapping amplitudes (>200 nm) are mechanically possible. Another interesting avenue to explore would be the application of multi-color s-SNOM imaging to simultaneously record the response at several wavelengths of the sample in liquid undergoing dynamic processes such as neurotoxic protein aggregation[49].

We believe that the accuracy and versatility of our method open up unprecedented avenues for future studies of even more complex lipid-associated phenomena. Furthermore, our findings lay the groundwork for understanding, cooperative and non-linear dynamics in nanoscale systems offering exciting opportunities for further exploration of a multitude of other highly relevant systems, ranging from dynamic metal-organic-frameworks in chemistry[39,65], the assembly of Alzheimer-associated peptides in medicine[66], to the electronic modulation of 2D materials in physics[52,67].

## Methods

### Near-field optical microscopy setup for in-situ photoswitching

The measurements were performed using a commercially available s-SNOM in reflection mode (neaSCOPE, *attocube systems*, Haar, Germany), which records correlated AFM-topographic and optical near-field measurements. The AFM tip operates on the upper surface of a stretched 10 nm-thin SiN membrane, which protects nano-scale objects adsorbed to its underside thus preserving their native environment[18]. Below the membrane an approximately 100 to 150 nm wide sample area is probed, given by the tip-limited spatial resolution[18]. All experiments were conducted with PtIr-coated AFM tips with a tip radius of 60 nm used as a scattering probe (nanoFTIR-tips, *attocube systems*, Haar, Germany) operated in tapping mode with a tapping amplitude of 80 nm a tapping frequency $\Omega$ of approximately 250 kHz at about 85% of the resonance frequency. In principle, choosing tapping amplitudes between 60 and 80 nm reduces deformation of the membrane by the tapping tip and is a good compromise between SNR and a background free signal[30]. A detailed description of the fundamental working principle of s-SNOM is, e.g., provided in the following literature reviews[12,13]. In short, the optical signal from the tip is attained by focusing laser light onto the tip and collimating the backscattering of the scanning probe via a parabolic mirror. Subsequently, the backscattered light is detected by a nitrogen-cooled mercury cadmium telluride (MCT) detector (IR-20-00103, *Infrared Associates Inc.*, Stuart, USA), which is sampled by the internal data acquisition card (DAQ) at a high enough bandwidth to demodulate the recorded signal at harmonics $n\Omega$ of the tapping frequency $\Omega$ by the DAQ electronics in order to separate the near-field signal from unwanted far-field signals. The near-field optical amplitude and phase images and the near-field signal traces at a defined tip position are recorded with a pseudo-heterodyne interferometric detection scheme[68], using an optical parametric oscillator laser (OPO) with a 1050 nm pump laser and an ultra-broadly tunable MIR output ranging from $\lambda = 1.4\,\mu m$ to $\lambda = 16\,\mu m$ attained by difference frequency generation (DFG) in a non-linear crystal (Alpha with MIR extension, *Stuttgart Instruments*, Stuttgart, Germany). The *Stuttgart Instrument* laser works with a pulse rate of 40 MHz and a temporal pulse duration of 400 fs. The resonance-specific IR images are achieved by using a grating monochromator to limit the spectral bandwidth to 10 cm$^{-1}$ and tuning the laser to the desired wavelength of the molecular absorption. The output power of the laser is attenuated by a wire grating (*Lasnix*, Berg, Germany) to about 3 mW before reaching the beamsplitter, which commonly understood to be in the optimal range for achieving a high SNR and good for avoiding tip and sample damage in Ps-Het measurements[68]. The pseudo-heterodyne detection allows for the decoupling of the demodulated complex-valued scattering coefficient $\sigma_n(\omega)$ into optical amplitude $s_n(\omega)$ and phase $\varphi_n(\omega)$ for a chosen wavelength[68]. The detected optical amplitude relates to the reflectivity and the optical phase to the IR absorption of the sample under the tip position[69]. The integration time $t_p$ of the Ps-Het based transient signal trace method defines the time interval at which the continuously read-out Ps-Het data points are averaged and the spacing of consecutive data points in time.

The nano-FTIR spectra and white light images were acquired with a broadband IR laser source based on difference frequency generation (DFG) (FFdichro_midIR_NEA 31002, *Toptica Photonics*, Martinsried, Germany)[27,70]. The nano-FTIR laser operates at 80 MHz repetition rate delivering 400 fs pulses. The output power was about 400 μW, and the covered spectral range between 1200 and 2000 cm$^{-1}$ (7.6−5 μm), achieving maximum power at 1666 cm$^{-1}$ (output mode D). The back-scattering from the tip was analyzed by an asymmetric Michelson interferometer with the tip and sample in one and a moveable reference mirror in the other arm of the interferometer. The detected signal was demodulated at harmonics n of the tapping frequency $n\Omega$ and Fourier-transformed to obtain spectra of the optical amplitude $s_n$ and phase $\varphi_n$, which are referenced to spectra recorded on a clean silicon surface to eliminate atmospheric and instrumental artifacts in the measured spectra[27].

The nano-FTIR spectra shown in Fig. 1 were recorded with an interferometer scan length of 350 μm, resulting in a nominal spectral resolution of 14.3 cm$^{-1}$. During the interferometer movement, 1200 discrete data points were recorded with an integration time of 35 ms per point. The shown spectra are an average of 20 nano-FTIR spectra recorded consecutively resulting in a total acquisition time of 14 min for one spectrum shown in Fig. 1f. Importantly, both averaged spectra in Fig. 1f have been recorded consecutively to minimize the time difference between the spectra resulting in unwanted outside influence on the spectroscopic response of the sample. The nano-FTIR spectra were recorded with such a relatively low spectral resolution as the SNR of a FTIR spectrum and the spectral resolution correlate inversely[9] and for liquid s-SNOM measurement on a liquid sample fast measurement with high SNR is essential. We choose to plot the optical phase $\varphi_n$ as opposed to the sometimes utilized imaginary part of the near-field signal as it correlates linearly with the infrared absorption of the sample in contrast to the imaginary part[41]. The near-field phase spectra are best compared to far-field ATR-FTIR spectra since ATR-FTIR is a technique that uses the penetration of evanescent waves through an interface to probe the sample, and thus the recorded ATR-FTIR spectra closely resemble the tip-enhanced near-field phase spectra. For white light imaging, as displayed in Fig. 1e and used to identify the vesicles for spectroscopy with the broad band nano-FTIR laser, the reference mirror of the asymmetric Michelson interferometer is fixed at the position with a zero optical path difference between the reference and tip-scattered beams. This results in a maximally constructive interference of all spectral components of the nano-FTIR output spectrum and a maximal optical amplitude $s_n$. The resulting white-light signal represents the backscattering of the sample below the tip averaged

over the whole laser output spectrum weighted by the laser power and the detector and setup responsivities, or in other words, a spectrally averaged near-field signal.

The photoswitching was performed by alternating between two diodes emitting at 365 nm and 465 nm wavelength, respectively (*Prizmatix*, Holon, Israel). The illumination power was set to around 30 mW (the maximal output power of the diodes), and the beam was concentrated by a section of the parabolic mirror as sketched in Fig. 1b, ensuring that the SiN membrane around the tip was homogeneously irradiated over an area of around $0.5 \times 0.5 \, \text{mm}^2$. This yields an irradiance of $E = 12000 \, \frac{\text{mW}}{\text{cm}^2}$. The illumination parameters were kept constant for all photoswitching experiments in the manuscript.

## Liquid s-SNOM sample preparation

Liquid s-SNOM samples were prepared by first pretreating the cavity side of a 10 nm thin and $250 \times 250 \, \mu\text{m}$ wide SiN membrane pretensioned on a commercial silicon chip (NX5025Z, *Norcada*, Edmonton, Canada) with a 30 min UV-Ozone cleaning procedure to increase the wettability of the membrane, then mounting the chip on a custom-made membrane holder and drop-casting 15 μL of the sample into the cavity side of the membrane[18]. After several minutes of incubation, the sample holder was sealed to prevent the evaporation of water, and placed in the s-SNOM system.

## ATR-FTIR spectroscopy

ATR-FTIR spectra were recorded with a commercial spectrometer (Frontier™ FT-IR spectrometer, *PerkinElmer*, Waltham, USA) with a proprietary KBr beamsplitter and a diamond ATR crystal. The spectra were recorded with a wavenumber range from 650 cm$^{-1}$ to 4000 cm$^{-1}$ with a spectral resolution of 4 cm$^{-1}$ and averaging over 60 scans.

## Finite dipole model for s-SNOM

Analytical calculations of near-field response were conducted using Python. To predict the optical response of a multi-layered sample, the finite dipole model of near-field interaction of a metal ellipsoid with a nearby material was extended to multiple layers via the transfer matrix method[18,71]. Results are shown in Fig. S3, using the modeling parameters $a = 80 \, \text{nm}$ (tapping amplitude), $r = 60 \, \text{nm}$ (tip radius), $L = 300 \, \text{nm}$ (length of spheroid), $g = 0.7e^{i0.06}$ (charge induced in the tip).

## Preparation of Liposomes

The photolipid azo-PC was synthesized according to a previous protocol[6]. Vesicles were prepared by electroformation using a vesicle prep pro device (*Nanion Technologies*, Munich, Germany). Two conductive glass substrates (coated by Indium tin Oxide, ITO) were separated by an O-ring, forming a sandwich-constructed chamber. Lipids were dissolved in chloroform at a concentration of 10 mM. 20 μL of the lipid solution (50% DOPC (*Merck*, Darmstadt, Germany) and 50% azo-PC) were added on the conductive side of ITO substrate within the O-ring. After evaporation of the chloroform, the O-ring chamber was filled with 250 μL of sucrose solution at 300 mM concentration. Sucrose solutions were prepared with $H_2O$ or $D_2O$. Electroformation was conducted by applying an electric field (5 Hz, 3 V) at 37 °C for 120 min.

## Reporting summary

Further information on research design is available in the Nature Portfolio Reporting Summary linked to this article.

# Data availability

The main data supporting the findings of this study are available within the article and its Supplementary Information files. Additional data that support the findings of this study are available via Zenodo at https://zenodo.org/records/15584835.

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

## Acknowledgements

The authors would like to thank J. Rädler and B. Nickel for valuable discussions. This project was funded by the Deutsche Forschungsgemeinschaft (DFG, German Research Foundation) under grant number TI 1063/1 (Emmy Noether Program) and the Center for NanoScience (CeNS). Funded by the European Union (ERC, METANEXT, 101078018 and EIC, NEHO, 101046329). Views and opinions expressed are however those of the author(s) only and do not necessarily reflect those of the European Union, the European Research Council Executive Agency, or the European Innovation Council and SMEs Executive Agency (EISMEA). Neither the European Union nor the granting authority can be held responsible for them. TL was supported by the Deutsche Forschungsgemeinschaft (DFG) through the Collaborative Research Center SFB1032 (Project No. 201269156, Project A8). J.Z. is supported by the China Scholarship Council. S.A.M. additionally acknowledges the Lee-Lucas Chair in Physics.

## Author contributions

T.G., F.K., T.L., and A.T. conceived the idea and planned the project. T.G., E.B., and K.K. performed the near-field and far-field measurements. J.Z. and D.T. provided and prepared the photoswitchable lipid sample. E.B. performed analytical simulations. T.G., E.B., F.K., J.Z., T.L., and A.T. helped with analyzing the experimental data. T.G., F.K., and A.T. wrote the manuscript with input from all the authors. A.T., T.L., F.K., and S.A.M. managed and supervised various aspects of the project.

## Funding

## Competing interests

F. Keilmann is a scientific advisor to attocube systems AG, manufacturer of the s-SNOM used in this study. T. Gölz obtained financial support for his PhD thesis from attocube systems AG. K. Kaltenecker is employee of attocube systems AG. The remaining authors declare no competing interests.
