## [Peer Review file · Nature Communications]

Transient infrared nanoscopy resolves the millisecond photoswitching dynamics of single lipid vesicles in water

Corresponding Author: Professor Andreas Tittl

Version 0:

Reviewer comments:

Reviewer #1

(Remarks to the Author)

In their manuscript, T. Golz et al. describe nanoscale imaging and spectroscopy of photo-switchable liposomes. The authors use one of several recently developed approaches for IR s-SNOM imaging at liquid interfaces, namely imaging through a transparent SiN membrane. They also incorporate a separate visible light source for activating the photo-switchable azobenzene-based lipids in situ. The authors image the shape of the lipid vesicles using the optical IR s-SNOM signal. They find that the shape of the adhered lipid vesicles becomes less circular after photo-switching. While the results are technically impressive, the analysis could be improved. Several conclusions need more supporting analysis and substantiation. Some experimental details are insufficiently explained. This manuscript may be suitable for publication in Nature Communications following revisions. Specific points below:

1. The authors do not clearly state their spectral resolution for nano-FTIR measurements. The spectra shown appear to be highly processed. The highly processed appearance of these spectra may be a result of low-resolution data collection combined with specifics of their Fourier Transform algorithm.

2. The authors do not provide sufficient explanation of the optical phenomena and methods for the reader to have a physical understanding of the meaning of “white light” images.

3. The authors do not explain signal-to-noise of their phase spectra, so it is difficult to discern a peak versus noise in the spectrum.

4. There appear to be no error bars in this manuscript. As an example: in Figure 2d, what methods were used to extract the full width at half maxima? Is this a fit of some sort? Please provide error bars.

5. Please provide scale bars for the images in Figure 3c. Please explain the boundary criterion “ $s_2=7.8$ a.u.” There is an extra space after the word circularity.

4. The authors claim a sigmoidal rather than exponential response in the data of Figure 4c and 4d, yet they do not show any fits. The authors also speculate that the asserted sigmoidal shape relates to a cooperative behavior. The authors do not include sufficient explanation or modeling to justify the claim of “cooperative effects between the lipid molecules.”

I have several questions:

-How repeatable are the time traces in Figure 4 following photo-switching?

-Is the “delayed-onset” consistent each time when repeating the same measurement?

-The illumination intensity at the focus spot is not provided for either 365 nm or 465 nm light. What was the intensity and was this constant for all measurements?

-Did the authors consider varying the intensity of illumination?

-The authors declare a time constant of 500 ms. How is this defined? If for example, 500 ms represents the time constant of a lock in, an exponential decay will be convoluted with the averaging time used in the lock in, contributing to a sigmoidal appearance.

5. Scale bars appear to be missing labels in Figure 4

6. Is circularity defined or used anywhere in the scientific literature, or is this a purely ad hoc fit?

7. The references section includes egregious self-citation. Approximately 30% of all references are self-citations, including references for IR s-SNOM development [12,14,15, 16, 25,27, 28,42,43,44,48,49,50,51], authored by Keilmann, Hillenbrand, or both. The review paper [13] appears to be the only IR s-SNOM citation to another group. Is IR s-SNOM a niche tool only used by Hillenbrand and Keilmann? Did they develop IR s-SNOM methods entirely independent of the rest of the scientific community? Are they the only researchers to apply this to molecular or biological systems or systems involving a liquid interface?

Reviewer #2

(Remarks to the Author)

The manuscript entitled "Transient infrared nanoscopy resolves the millisecond photoswitching dynamics of single lipid vesicles in water" offers an experimental investigation into the resolution of lipid vesicle photoswitching dynamics, which is relevant to advancements in nanomedicine, photopharmacology, and drug delivery systems. The authors employ transient infrared nanoscopy, clearly describing the method and its ability to achieve the spatio-temporal resolution (150 nm-30 ms), specifically focusing on photoswitchable liposomes within a liquid cell. This methodological approach significantly contributes to the practical application of nanoscopy in liquid experiments. Overall, the study is well-conducted, presenting a novel approach to examining lipid vesicle dynamics through transient infrared nanoscopy. The article demonstrates a degree of innovation and is likely to attract interest from researchers within this domain. I recommend its publication in Nature Communications following several minor revisions.

1. In Figures 2 and S3, regarding the flattening of the lipid nanovesicles, could the authors provide the complete topography image rather than only the line cut? I am particularly interested in the overall shape of the flattened lipid nanovesicle on the SiN membrane.

2. The abstract and section title claim a temporal resolution of 50 ms, as stated in the phrase "50 ms resolved MIR near-field signal tracking of the photoswitching dynamics of a single lipid vesicle." However, the main text and Figure S8 present results at 30 ms, which show a 3.6 sigma, exceeding the temporal resolution limit. This discrepancy should be addressed.

3. As the lateral resolution of 150 nm - 200 nm is relatively large for typical nanoscopy in liquid ["Amplitude- and Phase-Resolved Infrared Nanoimaging and Nanospectroscopy of Polaritons in a Liquid Environment", Nano Lett. 2021, 21, 1360–1367; "In Liquid Infrared Scattering Scanning Near-Field Optical Microscopy for Chemical and Biological Nanoimaging", Nano Lett. 2020, 20, 4497–4504; "Liquid-Phase Peak Force Infrared Microscopy for Chemical Nanoimaging and Spectroscopy", Analytical Chemistry 2021 93 (7), 3567-3575; "High-sensitivity infrared vibrational nanospectroscopy in water", Light: Science & Applications (2017) 6, e17096; "Antenna array-enhanced attenuated total reflection IR analysis in an aqueous solution", Nanoscale, 2019,11, 18543-18549], which claim typical spatial resolution of a few tens of nm. Moreover the liquid cell does not allow for the observation of the complete shape of the lipid nanovesicle. The authors should discuss potential methods for overcoming these limitations.

Reviewer #3

(Remarks to the Author)

Gözl et al. demonstrated a novel transient MIR nanoscopy method for investigating photoswitching dynamics in situ on nanocarriers and showcase this approach on individual photo-switchable lipid nanocarriers (lipid vesicles) isomerisation at the single liposome level in aqueous environments. Golz and co-authors present the first s-SNOM in situ study on actively photo-induced nanocarrier structure-property relationship change modulated by ultraviolet and blue light illumination at a temporal resolution tolerant for practical applications (50 ms, can be considered as being real time in many cases).

This manuscript is well-written and easy to follow in a clear structure based on well-thought experimental design and convincing s-SNOM experimental results. The results have demonstrated the unique advantage of MIR s-SNOM over typical ATR-FTIR and far-field spectroscopy approaches. This remarkable study is a step changer for near-field optics and nanophotonics based the group's solid experimental studies published before. The demonstrated approach in this manuscript is a milestone towards studying other unlabelled soft matter complex dynamics under native physiological conditions via light-matter interactions in the future.

The nanoscopy approach in this manuscript will be a new research paradigm for revealing the microscopic origin on fundamental biological and chemical processes in native liquid environment by utilising nanoscale light-matter interactions, offering unparalleled insights for material/chemical scientists as well as microbiologists in future studies beyond lipid systems.

I recommend a minor revision and the authors are expected to address my comments below before publishing on Nature Communications.

Comments:

1. [Relevant to the liquid cell] The vesicles are attached at the bottom surface of SiN membrane via van der Waals (vdW) forces, which may be considered as not as native as free running molecules in liquid. In some certain chemical reactions, this externally introduced vdW forces may modulate microscopic biological or chemical process (since the vesicles may be flattened or deformed when adhering to the SiN membrane). Also, the SiN may be reactive to attached biomolecules or

chemical compound if we want to study at the single molecular level. Are we still probing deformed molecules, which is not as native, in liquid environment? It will be helpful to clarify for non-s-SNOM experts: (1) is there a strategy to mitigate this issue or (2) do we need to find application scenarios to study chemical process not perturbed by nanoscale adhesion (or study problems directly relevant to nano-adhesion)?

Will other 10-nm thin film other than SiN be able to serve as capping layer for control experiments to rule out potential artefacts introduced by the (SiN) membrane or by vdW forces introduced by capping layers itself?

What kind of requirements (e.g., dielectric and mechanical properties) of the capping layer required for this s-SNOM approach? For the wavelength range outside of nano-FTIR, is there other candidates of capping layers? For example, being transparent (dispersion-less) to both pumping & probe radiation wavelength but sensitive to the interrogated sample solely?

2. The authors deliberately mis-align two LEDs (365 nm & 465 nm) off the tip to avoid creating near-fields at non-IR wavelengths. If this two LEDs were aligned at the tip simultaneously, will the reported results of photo-switchable behaviour / nano-FTIR spectral results change? The authors could consider adding these results as control experimental results. This could create motivation in s-SNOM community for another multi-colour s-SNOM modality spanning from THz to UV for transient IR nanoscopy or even transient THz nanoscopy for using this kind of liquid cell.

3. The authors showed 1-nm membrane protrusion (mechanical footprint) with pronounced optical signals due to vesicles from nano-FTIR confirmation. For non-trivial topography scanned by s-SNOM, is there a way to remove potential spurious optical signal contribution not relevant to the interrogated sample itself, especially when multiple high harmonics are not possible? This could be crucial when we heat up the liquid cell add solvents (control PH) to trigger other biochemical processes.

Discussion on these will be helpful for readers when future researchers adopting this liquid cell for experimental design or following this method to prepare customised membranes / liquid cells, which does not have reproducible flatness as a commercial membrane.

4. For MIR optical phase raw images collected by the tuneable monochromatic laser (e.g., Fig. 3b, Fig. 4b/f, or Fig. S4), I can observe non-negligible non-zero phase signals at pixels not overlapping with optical amplitude (which we consider they are correlating to sample flattened topography attached below the membrane). Is there other interpretation other than background noise on them (e.g., Fig.4b and Fig. S4 4th column counting from the left to right hand side)?

5. Since we cannot measurement directly the sample topography in this exciting liquid cell achievement, if researchers would like to track photo-release bio-chemical process, e.g., drug delivery, will the current approach (or revised approach potentially) be able to track this free running process (not attached to the membrane) by tuning the probing depth with tip tapping amplitude potentially? How shall we be confident what are we measuring below (but not attached to) the membrane bottom surface?

6. How will the tip approach curve look like at the spatial position with vesicles and with the sucrose solution (D2O)? Can the authors therefore use finite dipole model to predict the probing depth for the liquid cell (considering the 10-nm Si membrane)?

7. Have the authors explored the largest tapping amplitude suitable for this kind of in situ s-SNOM measurements using liquid cell? For certain wavelength like THz nanoscopy, the usually adopted tapping amplitude could be > 200 nm or 300 nm for measurable tip-scattered signals. Will a 100-nm SiN membrane be mechanically strong enough? This kind of information will be useful for translating this approach to studies in applications for a broad readership.

8. I noticed the g parameter in finite dipole model is 0.6 as a real number, will the used parameters be able to recover the measured approach curve at the monochromatic frequency?

9. In many nano-FTIR literature, researchers may use the imaginary part of normalised s-SNOM signal to locate the spectral absorption peaks. The authors chose to use phase signals directly in this manuscript for assignment of functional group, any specific reason? A more reasonable spectral peak position for functional group assignment?

10. Since the resonance peak intensity decrease at 1606 cm^{-1} is the indication of trans-to-cis isomerisation after 365-nm illumination, it is better to highlight how long the whole measurement took (from topping up liquid nitrogen to finish the nanospectroscopy) as future benchmark. The MIR phase and amplitude intensity decrease could be perceived mistakenly as correlating to MCT detector cooling performance.

11. The author used a wire grating to attenuate the output power (tuneable laser) to about 3 mW before reaching the beam splitter, what is the reason?

12. [p4, line 114] It is better to describe trans & cis for non-Latin readers without chemical background. Fig. 1d clarifies cis-trans isomerism [p4, line 113] visually --- consider adding one sentence to describe these two in the text.

13. Fig. S2a is blurry. It is hard to see the functional group in purple circle, considering the vector figure. Minor: the red colour for carbon chain does not match with the colour in panel b & c (Fig. S2).

Reviewer #4

(Remarks to the Author)

The manuscript reports experimental approach seeking to probe the morphology and dynamics of a phospholipid vesicle containing photoswitchable azo-lipids. They performed in-situ s-SNOM studies on actively switched vesicles, by repeatedly and reproducibly switching between cis and trans states using UV/blue lights, with a spatio-temporal resolution of 100-150 nm and 50 ms. These studies are useful for understanding the photo-induced drug release processes at the cellular level. s-SNOM is a powerful and label-free surface sensitive technique for surface characterizations. The technique involves an AFM tip oscillating vertically above the sample surface and a focused pulsed/CW laser scattering from the apex of the tip. The scattered beam of light contains information about near-field interactions between the sample and the tip, thereby leading to a high spatial resolution. They performed in-situ photo-induced dynamic measurements on single lipid vesicle level.

Although the results obtained are partially insightful, the manuscript fails to substantiate the primary claims as I have described below.

In the abstract, the authors claim that their study leverages chemical recognition which I agree only partially. I do not see a significant difference in the spectrum shown in Figure S1 between the cis and trans states of the azo-PC. Here, the authors rely solely on the change in intensity. However, their previous measurements (Crea et al., *Front. Mol. Biosci.*, 9, 905306, 2022) demonstrated that the trans states show breathing modes at 1603 and 1580 cm^{-1} whereas the cis states show at 1511 and 1496 cm^{-1} which are very distinct and easily identifiable.

Applicability to real systems. The system and the chemical environment chosen here will have poor applicability in the real systems because of (1) using D2O instead of H2O or a buffer because buffer can alter the photoswitching process, however, I agree that H2O hinders the ring breathing modes in the spectra, (2) the 50% ratio of the azo-lipid seems too high. Authors should explain why they have considered 50% of azo-PC in their study instead of a smaller ratio which could have been more realistic. To my understanding at 50:50 ratio the vesicle could be unstable and vulnerable to any foreign molecule it might encounter during the drug delivery process and hence loses its applicability. Measurement of the efficiency of drug release upon switching could have demonstrated the potential application of this system.

It is disappointing to see that the materials and method section does not properly describe the laser parameters, e.g, whether it is a continuous wave laser or a pulsed laser, the pulse length and frequency if it is a pulsed laser. Also, the manuscript fails to describe the underlying theory of the sSNOM method which is the primary focus of the entire manuscript. If it is a pulsed laser, the usual practice is to set the frequency of the laser pulse to be more than four or six times the frequency of the AFM tip depending on whether the 2nd order or 3rd order harmonic demodulations have been considered (see *Nat. Commun.* 7, 13212, 2016 by Wang et al.). Authors need to carefully address this issue.

Also, the thermodynamic characterizations (such as phase transition) of this vesicle has not been done, so the phase of the system is completely unknown. Since the switching process highly depends on the lipid phase it is crucial to characterize the thermodynamics of the vesicle prior to the photoswitching study.

There are already some nice literatures available in the similar direction. Such as by Quaroni et al. *RSC Advances* 8, 2786 (2018), and by Cernecu et al., *Analytical Chemistry* 90, 10179 (2018), and by Roman et al., *Nanotechnology* 30, 425502 (2019). They nicely described the spectroscopy and/or imaging data from lipid droplets, vesicles and also from the single cell. Authors should describe the speciality of their method used in this study and also, if possible, should complement their results with some other methods (e.g., Small angle X-ray scattering).

Page 3. Authors described the advantages of using SiN membrane. But the possible disadvantages, such as the potential hindrance of the near-field interactions by SiN, have not been discussed.

Page 10, Figure 3. Whether any sort of corrections have been applied on the obtained images for enhancing the contrast should be described. Also, according to my understanding, the synchronization of detector sampling rate with the pulse rate of the laser source plays a great role to maintain the near-field contrast. Did the author consider these points?

Page 11. The statement "switching dynamics at a modest SNR of about 4 down to around 50 ms (Figs. S8, S9, S10)". Figs. S8, S9, S10 suggest that the switching dynamics are a few tens of seconds long whereas it is claimed that the time resolution is less than 50 ms. Do they mean temporal resolution as the step size used in the measurement? Do the authors have any reference measurements of the temporal resolution? Also, since the entire result and the subsequent claims are primarily based on the switching dynamics, I suggest to fit the data shown in Figs. S8 and S9 with a transient equation, and to move them to the main manuscript with the discussions of the findings.

Version 1:

Reviewer comments:

Reviewer #1

(Remarks to the Author)

The authors have sufficiently addressed many of my concerns.

The authors have not yet addressed my concerns about the scientific interpretation of their time-resolved measurements. The authors use a sigmoid equation lacking citation. The authors claim without evidence or sufficient citation that the

appearance of such a sigmoid in time-resolved data justifies a claim of cooperative photoswitching.

The authors have also not yet addressed my concerns about their experimental time resolution.

Pending these minor corrections, this manuscript will be suitable for publication in Nature Communications.

Detailed below related to my original review items 1 and 6:

1:

Thank you for the clarification. I believe the scan length reported in the author response (350 μm) is correct, but the scan length reported in the edited manuscript (300 μm) is a typographical error. If the scan length is 350 μm , then I believe the spectral resolution is $1/0.07=14.2857 \text{ cm}^{-1}$, which rounds to 14.3 cm^{-1} not 14.2 cm^{-1} as written.

6:

I believe in their response, the authors erroneously reported the illumination as 0.48 mW m^{-2} . I believe the value should be 0.48 MW m^{-2} . I encourage the authors to confirm the correct value and include this information in the revised manuscript or supplement.

I believe the word sinusoidal in the author response was a typographical error and the authors intended to write sigmoidal.

The authors state in their reply "The time constant of 500 ms is based on a 500 ms average of the continuously read-out Psu-Het signal" and then claim "the recording switching dynamics is not convoluted as it is much slower." Let me be clear: observed dynamics are necessarily convoluted with instrument response time in ANY experiment. I would argue strongly that an instrumental time resolution of 0.5 seconds will absolutely affect the value of tau, which is reported in the manuscript as "ranging from 0.7 to 3.9s." Furthermore, depending on data processing, a 500 ms running average could result in an instrument resolution closer to 1 s, depending on implementation of the moving average. The authors must absolutely address experimental uncertainty before making claims based upon their fits.

The authors state in the original manuscript and in their reply that a sigmoidal is indicative of a cooperative response. The authors need to provide a citation clearly justifying the connection between a sigmoid and a cooperative response. I have no doubt that the photoswitching is related to a first order phase transition, but I do not see the connection specifically to the mathematical function of a sigmoid. (If the switching is associated with a first order phase transition, then I naively would expect a variable induction time, which the authors fit include in their sigmoid as term t_d , but I do not see an obvious connection to the shape of the sigmoid and the fitted tau parameter.) The authors must include a citation clearly justifying the use of the sigmoid function and explain how the appearance of a sigmoidal justifies claims of cooperative behavior before drawing scientific conclusions from their fit parameters.

Reviewer #2

(Remarks to the Author)

The rebuttals from the authors are clearly written and address all my requests and comments. As the paper reports important experimental findings in the field of the "Transient infrared nanoscopy for biomaterials in liquid", I recommend the manuscript for the publication of Nature Communications.

Reviewer #3

(Remarks to the Author)

The authors have addressed my comments in detail. I can recommend this manuscript to be published on Nature Communications, pending the following minor comments for the benefit of readers:

1. 30 ms temporal resolution --- as Reviewer #4 pointed out, this could be misleading, as readers may incorrectly interpret it as indicating actual signal changes, which in reality happen between 1 to 2 seconds.

To clarify, perhaps on p.13 or p.11, the authors could add a simple sentence explicitly stating that this refers to the time per sampling step required to obtain the time-domain interferogram in nano-FTIR measurements. Also, note that p.19, line 544 states "35 ms per point".

2. As the authors pointed out in their response letter, finite dipole model is less useful for complex unknown biological samples. It will be interesting to discuss alternative approaches to mitigate this. I would like to refer the authors to a relevant overlooked review paper on THz nanoscopy (Appl. Phys. Rev. 11, 021306 (2024) <https://doi.org/10.1063/5.0189061>), which proposes an envisioned measurement procedure (Fig. 12 in that reference) in light of Sci. Rep. 11, 21860 (2021).

3. Since s-SNOM probes local dielectric constants, it would be helpful for non-chemistry readers if the authors could comment on how relative changes in observed local dielectric constants may be interpreted in terms of biochemical processes in liquid environments.

Version 2:

Reviewer comments:

Reviewer #1

(Remarks to the Author)

The authors have addressed my comments. I recommend this manuscript for publication in Nature Communications.

Response letter to reviewer comments

Reviewer #1 (Remarks to the Author):

General statement:

In their manuscript, T. Golz et al. describe nanoscale imaging and spectroscopy of photo-switchable liposomes. The authors use one of several recently developed approach for IR s-SNOM imaging at liquid interfaces, namely imaging through a transparent SiN membrane. They also incorporate a separate visible light source for activating the photo-switchable azobenzene-based lipids in situ. The authors image the shape of the lipid vesicles using the optical IR s-SNOM signal. They find that the shape of the adhered lipid vesicles becomes less circular after photo-switching. While the results are technically impressive, the analysis could be improved. Several conclusions need more supporting analysis and substantiation. Some experimental details are insufficiently explained. This manuscript may be suitable for publication in Nature Communications following revisions.

Our response:

We thank the reviewer for the kind appreciation of our work and hope the following response to the comments will help to settle the stated issues.

Reviewer comment:

1. The authors do not clearly state their spectral resolution for nano-FTIR measurements. The spectra shown appear to be highly processed. The highly processed appearance of these spectra may be a result of low-resolution data collection combined with specifics of their Fourier Transform algorithm.

Our response:

The reviewer is correct with the intuition that the smooth appearance of the spectra is due to a low resolution of the nano-FTIR spectra. We recorded the nano-FTIR spectra with an interferometric distance of 350 μm resulting in a spectral resolution of 14.2 cm^{-1} . Over the 350 μm distance 1200 data points have been recorded with an integration time of 35 ms for each data point. The displayed spectra in **Figure 1f** are an average of 20 single nano-FTIR spectra. The averaging of FTIR spectra is a common process in nano-FTIR and increases the SNR of the spectra. We choose the low resolution due to the inverse correlation between SNR of the recorded nano-FTIR spectra and the spectral resolution. As the liquid-lipid-membrane system is quite delicate and the lipid particle can move over time especially during photoswitching, we opted for nano-FTIR spectroscopy with lower spectral resolution but faster acquisition time. Importantly, the spectral resolution is still sufficient to resolve the main switching peak located around 1606 cm^{-1} .

Actions taken:

We added the following part in the Materials and Methods section of the revised manuscript on **page 19** to clarify the spectral resolution:

"The nano-FTIR spectra were recorded with an interferometer scan length of 300 μm , resulting in a nominal spectral resolution of 14.2 cm^{-1} . During the interferometer movement, 1200 discrete data points were recorded with an integration time of 35 ms per point. The shown spectra are an average of 20 nano-FTIR spectra recorded consecutively resulting in a total acquisition time of 14 min for one spectrum shown in **Fig. 1f**. Importantly, both averaged spectra in **Fig. 1f** have been recorded consecutively to minimize the time difference between the spectra resulting in unwanted outside influence on the spectroscopic response of the sample. The nano-FTIR spectra were recorded with such a relatively low spectral resolution as the SNR of a FTIR spectrum and the spectral resolution correlate inversely⁹ and for liquid s-SNOM measurement on a liquid sample fast measurement with high SNR is essential. We choose to plot the optical phase ϕ_n as opposed to the sometimes utilized imaginary part of the near-field signal as it correlates linearly with the infrared absorption of the sample in contrast to the imaginary part⁶⁶.

Reviewer comment:

2. The authors do not provide sufficient explanation of the optical phenomena and methods for the reader to have a physical understanding of the meaning of “white light” images.

Our response:

The reviewer is correct that we did not provide sufficient explanation of the white-light imaging microscopy mode as the concept behind this imaging mode is not straightforward.

Actions taken:

We added on **page 6** and **in the caption to Figure 1e** that the spectrally average MIR near-field amplitude image is a white light image to resolve possible confusion between the two terms. We also modified the label of the color bar in **Figure 1e** as shown below with a new label now reading spectrally average (white light) MIR signal s_2 (a.u.):

In addition, we have modified and extend the following paragraph in the materials and method section on white-light imaging on **page 19**:

"For The white light imaging, as displayed in **Fig. 1e** and used to identify the vesicles for spectroscopy in this study and placement of the tip at the proper position to record the nano-FTIR spectra subsequently with the broad band nano-FTIR laser, are obtained by placing the reference mirror of

the asymmetric Michelson interferometer is fixed at the position with a zero optical path difference between the reference and tip-scattered beams. This results in a maximally constructive interference of all spectral components of the nano-FTIR output spectrum and a maximal optical amplitude s_n . The resulting white-light signal represents the backscattering of the sample below the tip averaged over the broad whole laser output spectrum weighted by the laser power and the detector and setup responsivities, or in other words, a spectrally averaged near-field signal. The white-light images are used to identify the vesicles in this study, and the tip is placed at the proper position to record the nano-FTIR spectra subsequently."

Reviewer comment:

3. The authors do not explain signal-to-noise of their phase spectra, so it is difficult to discern a peak versus noise in the spectrum.

Our response:

We thank the reviewer for being interested in the SNR characteristics of the spectra. As stated above the shown spectra are an average of 20 single nano-FTIR spectra recorded on the shown single vesicle. Unfortunately, the commercial control software of the attocube microscope automatically averages the spectra and does not give access to the individual spectra. So, it is not possible for us to calculate error bars for the displayed spectra. This is an underlying limitation of the system that affects all users. However, the spectra show the clear characteristic peaks of the lipids as the comparison to **Fig. S1** demonstrates. Furthermore, the peaks that should be influenced by the photoswitching behavior are the only peaks that change in intensity. The carbonyl peak at 1742 cm^{-1} remains unaffected by the switching behavior and acts as an internal reference demonstrating that the spectral signal did not fluctuate between the two measurements and that the shown intensity changes indeed reflect a photoswitching event.

Actions taken:

We added the following sentence to results part of the manuscript on **page 5**:

" Importantly, the carbonyl signal at 1742 cm^{-1} in the nano-FTIR spectrum under 365 nm illumination does not change in intensity compared to 465 nm illumination, as is expected from the reference ATR-FTIR spectrum. This confirms that there was no change in laser power, optical tip alignment, or detector responsivity between the recording of the two spectra, which could lead to intensity modulations. "

As state above we expanded the materials and methods section on the nano-FTIR spectroscopy.

Reviewer comment:

4. There appear to be no error bars in this manuscript. As an example: in Figure 2d, what methods were used to extract the full width at half maxima? Is this a fit of some sort? Please provide error bars.

Our response:

We agree that the error values of **Figure 2d** are missing. We fitted the extracted profile with a gaussian function to extract the FWHM and the error values are square root of the diagonal elements in the obtained covariance matrix. We added error bars in places where it was possible like **Figures S8, S9**. However, as discussed above it was not possible to add error bars to the nano-FTIR spectrum or the ATR-FTIR spectrum as the commercial instruments only provide us with the averaged spectra.

Furthermore, the time traces are based on a single pixel data point recording giving us no statistics about the error margin of a single time trace.

Actions taken:

We added the error values to **Figure 2d** and the following caption on **page 8**:

"Extracted profiles along lines drawn in (c), allowing to extract the full widths at half maxima (FWHM) of 660 ± 17 nm, 580 ± 23 nm, 260 ± 22 nm, and 176 ± 14 nm, determined by fitting a gaussian function to the extracted profile (not shown)."

Reviewer comment:

5. Please provide scale bars for the images in Figure 3c. Please explain the boundary criterion “ $s_2=7.8$ a.u.” There is an extra space after the word circularity.

Our response:

We thank the reviewer for the careful reading of **Figure 3**. The criterion of $s_{2,crit}=7.8$ is chosen to separate the particle from the background such that $s_{2,crit}\approx(1/e)*(\max(s_2)-s_{2,background})$, or when the amplitude reaches around 35% of its maximum value at the center of the particle. Considering a profile drawn from the edge of the image to the center where the amplitude reaches its highest point (shown in the Figure below), we find that $\max(s_2)\approx 9.1$ and $s_{2,background}\approx 7.2$, hence the chosen criterion.

We agree that we should add a scale bar for **Figure 3c**.

Actions taken:

We added the following explanatory sentence to the part where the boundary criterion is defined in the main part of the manuscript on **page 9**:

"This value was chosen to encompass the area where the optical amplitude is above $\frac{1}{e}$ of its maximum value at the particle centre after background subtraction."

We added scale bars for **Figure 3c** found on **page 10**:

Figure 3: Near-field imaging of the reversible photoswitching of a 500 nm diameter lipid vesicle. (a) Monochromatic MIR amplitude (s_2) and **(b)** phase images (ϕ_2) at 1603 cm^{-1} of a lipid vesicle in D_2O being reversibly photoswitched between the *trans*-state and the *cis*-state, scale bars 500 nm. **(c)** Illustrated extraction of a "circularity" value for a lipid vesicle defined as $4\pi A/p^2$, with circumference p and area A , where the boundary criterion was $s_2 = 7.8 \text{ a.u.}$, **scale bar 300 nm.** **(d)** Reversible change in circularity between the *trans*- (blue) and *cis*-state (violet) of the lipid vesicle over two switching cycles.

Reviewer comment:

6. The authors claim a sigmoidal rather than exponential response in the data of Figure 4c and 4d, yet they do not show any fits. The authors also speculate that the asserted sigmoidal shape relates to a cooperative behavior. The authors do not include sufficient explanation or modeling to justify the claim of "cooperative effects between the lipid molecules."

I have several questions:

- How repeatable are the time traces in Figure 4 following photo-switching?
- Is the "delayed-onset" consistent each time when repeating the same measurement?
- The illumination intensity at the focus spot is not provided for either 365 nm or 465 nm light. What was the intensity and was this constant for all measurements?
- Did the authors consider varying the intensity of illumination?
- The authors declare a time constant of 500 ms. How is this defined? If for example, 500 ms represents the time constant of a lock in, an exponential decay will be convoluted with the averaging time used in the lock in, contributing to a sigmoidal appearance.

Our response:

We thank the reviewer for being interested in the transient time traces. We agree that the time trace shown in **Figure 4** should be fitted with a sinusoidal function. Our fit shows that the *cis*-to-*trans*-switching direction is on average faster than the *trans*-to-*cis*-direction.

Regarding the Reviewer's question about cooperative effects, we apologize for not clearly connecting the discussion about lipid-lipid interactions with the observed slow onset of the switching dynamics. We have now revised our manuscript, to explain this in more detail.

a,b) We have reproduced the time traces with a "delayed-onset" on the same vesicle in a time frame of several hours added time trace in **Fig. S7** recorded later on the same day on the same vesicle. Furthermore, the fitting of the time trace shown as new **Fig. S5** and **S6** show delay onset times t_d in a range of 15.7 to 9s.

c) The illumination intensity was constant for all measurements in the manuscript. We utilized the upper limit of output power of the LEDs of 30 mW. It is difficult to have an exact estimate of the illumination intensity of the sample surface as the beam is defocused over an area larger than the whole SiN membrane. In addition, some of the UV-light will be absorbed by the mirror systems. We estimate the upper limit of the irradiance E of the sample surface to be around:

$$E = \frac{P}{A} = \frac{30 \text{ mW}}{250 \mu\text{m} \times 250 \mu\text{m}} = 0.48 \frac{\text{mW}}{\text{m}^2}$$

d.) We did not vary the illumination intensity as this study is mainly a proof of concept of the s-SNOM ability to resolve such fast dynamic processes in soft matter systems in liquid. For the study we used the upper limit of the output intensity of the LEDs. In general, decreasing the output power of the LEDs would result in a slower switching rate of the system. One of the goals of this publication is to show the maximal temporal resolution of the transient nanoscopy method. Therefore, it did not make sense for us to lower the output power of the LEDs. In addition, the final photostationary state of the lipid system will not be changed by the illumination intensity. Therefore, the overall signal behavior should be very similar with a change in the time dynamics of the switching being present when varying the illumination intensity.

e.) The time constant of 500 ms is based on a 500 ms average of the continuously read-out of the Ps-Het signal. This means that each data point represents an average value over 500 ms of the signal trace signal, and consecutive data points are spaced 500 ms apart in time. As the switching dynamics of the lipid vesicle is in the order of several seconds, as the fits show below, the recorded switching dynamics is not convoluted as it is much slower.

Actions taken:

We added a sigmoidal fit (red curve) of the time traces in **Figure 4**:

Figure 4: Resolving the photoswitching dynamics of a single 500 nm nanoscale lipid vesicle by millisecond-MIR near-field signal traces. (a) Optical near-field amplitude image normalized to the surrounding D₂O ($s_2/s_{2,D_2O}$) and (b) near-field phase image (φ_2), recorded at 1603 cm^{-1} with 2 min acquisition time, directly before starting the MIR signal trace acquisition at the position of the cross, scale bars 500 nm. The recorded amplitude ($s_2/s_{2,D_2O}$) in (c) together with the phase (φ_2) in (d) reveal the photoswitching dynamics of the eight switching events at a temporal resolution of $t_p = 500\text{ ms}$. The symbols above (c) in combination with the dashed vertical lines indicate when the illumination wavelength was switched to the value written above. The blue and violet coloring of the signal traces mark the switching light's wavelength at each point. The black curves are moving averages of the measured points, and the red curves are sigmoidal fits of the switching process of the form $f(t) = \frac{L}{1 + e^{-\frac{(t-t_d)}{\tau}}} + C$ as displayed in detail in Fig. S5 and S6. (e) Optical amplitude and (f) phase images taken directly after acquiring the near-field traces show that the vesicle remained stable in both position and signal strength, even after eight consecutive switching transitions, scale bars 500 nm.

Furthermore, we added a close-up of each switching event with the associated fit as Figure S5 (amplitude signal s_2) and Figure S6 (phase signal φ_2). The figures are accompanied by the respective delay time t_d and time constant τ for each switching event:

Figure S5: Sigmoidal fit of the transient amplitude signal time trace for investigating the photoswitching dynamic shown in Fig. 4. The extracted time constants τ determines the steepness of the switching behaviour describing how fast the system responds. The average τ -value for the *trans*-to-*cis* switching is 2.8 ± 1.1 s, whereas the τ -value for the *cis*-to-*trans* is 1.5 ± 0.5 s, indicating that the *cis*-to-*trans* switching process occurs faster. The delay time t_d , defined as the interval between the switching of the light and the inflection point of the fitted curve, is 11.8 ± 1.0 s for *trans*-to-*cis* switching and 13.0 ± 2.6 s for *cis*-to-*trans* switching. The fitting was performed with a sigmoidal function of the following form $f(t) = \frac{L}{1 + e^{-\frac{(t-t_d)}{\tau}}} + C$.

Figure S6: Sigmoidal fit of the transient phase signal time trace for investigating the photoswitching dynamic shown in Fig. 4. The extracted growth parameters τ determines the steepness of the switching behaviour describing how fast the system responds. The average τ -value for the *trans*-to-*cis* switching is 3.0 ± 0.6 s, whereas the τ -value for the *cis*-to-*trans* switching is 1.6 ± 0.8 s, further indicating that the *cis*-to-*trans*-switching occurs faster. The delay time t_d is 11.7 ± 1.9 s for *trans*-to-*cis* switching and 13.6 ± 2.7 s for *cis*-to-*trans* switching. The fitting was performed with the same sigmoidal function as in Figure S5.

We added the following sentences to the result part of the manuscript on **page 12** and **13** to state the finding of the above fits:

"When fitting the MIR signals of the switching events in **Figs. 4c, d** with a sigmoidal response function (**Figs. S5 and S6 and red curves in Fig. 4c, d**), characteristic delay times t_d ranging from 10.3 to 15.7 s and growth parameters τ ranging from 0.7 to 3.9s could be extracted."

"The extracted growth parameters τ through the sigmoidal fit of the time traces (**Fig. S5, S6**) indicate a higher switching speed in the *cis*-to-*trans* direction in comparison to the *trans*-to-*cis*-direction, in agreement with previous reports."

We revised our manuscript, now including a discussion about cooperative effects between lipid molecules. The following paragraph has been added to the manuscript text:

"Cooperative effects promoting stronger lipid-lipid interactions of azo-PC in the *trans*-state have been reported in the literature. For example, the azobenzene groups of the lipid tails align side-by-side in a bilayer setting, which promotes the formation of H-aggregates^{28,42}. Owing to their molecular conformation, *trans*-photolipids are also stacked more densely, which is reflected by a lower membrane diffusion coefficient of *trans*-azo-PC compared to its *cis*-form³⁷."

a,b) We added a second 500 ms time trace as **Fig. S7** to the manuscript to emphasize the repeatability of the time traces and the following sentence to the main part of the manuscript on **page 14**:

"In addition, the time traces can be reproduced with good quality on the same sample (**Fig. S7**)."

Figure S7: Near-field time trace showing the normalized optical near-field amplitude ($S_2/S_{2,D20}$) recorded with 500 ms resolution. The time trace was recorded on the same vesicle as in **Figure 4** reproducing the switching dynamics.

c.) We added the following sentence to the method section of the manuscript on **page 20** to emphasize that the LED intensity was constant for all photoswitching experiments:

"The illumination parameters were kept constant for all photoswitching experiments in the manuscript."

d.) We commented on the idea of varying the illumination intensity to change the switching time on **page 13**:

"In our experiments, the accelerated signal changes often occur between three or four data points separated by 0.5 s, i.e. within 1.5 or 2 s. An interesting perspective for future studies could be to reduce the illumination intensity, resulting in slower switching dynamics and thus allowing deeper insight into this phenomenon³²."

e.) We added the following sentence to the method section of the manuscript to better explain the time resolution of the transient signal traces on page 18:

"The integration time t_p of the Ps-Het based transient signal trace method defines the time interval at which the continuously read-out Ps-Het data points are averaged and the spacing of consecutive data points in time."

Reviewer comment:

7. Scale bars appear to be missing labels in Figure 4

Our response:

We thank the reviewer for pointing out that the label of the scale bar is missing in Figure 4.

Actions taken:

We added the following label to the caption of Figure 4 on page 12:

"(a) Optical near-field amplitude image normalized to the surrounding D₂O ($S_2/S_{2, D_2O}$) and (b) near-field phase image (φ_2) recorded at 1603 cm⁻¹ with 2 min acquisition time, directly before starting the MIR signal trace acquisition at the position of the cross, scale bars 500 nm."

"(e) Optical amplitude and (f) phase images taken directly after acquiring the near-field traces show that the vesicle remained stable in both position and signal strength, even after seven consecutive switching transitions, scale bars 500 nm."

Reviewer comment:

8. Is circularity defined or used anywhere in the scientific literature, or is this a purely ad hoc fit?

Our response:

We thank the reviewer for being interested in the circularity metric. It signifies the comparability of the shape to a perfect circle and signifies the degree of deviation from the ideal circle shape. When the circularity has a value of 1 the vesicle can be seen as perfectly circular. This metric is a commonly employed shape factor meaning a dimensionless quantity used in image analysis and microscopy that numerically describe the shape of particles.

Actions taken:

We agree that the shape factor needs to be better explained and added the following sentence in the results part of the manuscript on page 9.

"Circularity is a commonly employed shape factor in image analysis that numerically describes the comparability of the vesicle to a perfect circle. The vesicle resembles a perfect circle at a value of one and deviates from the ideal circle shape with a lower value. "

In addition, we cited the following papers showing two examples of the use of circularity as a metric in scientific literature.

Ritter, N. & Cooper, J. New Resolution Independent Measures of Circularity. *J Math Imaging Vis* 35, 117–127; 10.1007/s10851-009-0158-x (2009).

Takashimizu, Y. & Iiyoshi, M. New parameter of roundness R: circularity corrected by aspect ratio. *Prog. in Earth and Planet. Sci.* 3; 10.1186/s40645-015-0078-x (2016).

Reviewer comment:

9. The references section includes egregious self-citation. Approximately 30% of all references are self-citations, including references for IR s-SNOM development [12,14,15, 16, 25,27, 28,42,43,44,48,49,50,51], authored by Keilmann, Hillenbrand, or both. The review paper [13] appears to be the only IR s-SNOM citation to another group. Is IR s-SNOM a niche tool only used by Hillenbrand and Keilmann? Did they develop IR s-SNOM methods entirely independent of the rest of the scientific community? Are they the only researchers to apply this to molecular or biological systems or systems involving a liquid interface?

Our response:

We agree with the reviewer that it is important to accurately capture the diversity of the scientific community working in the field of IR s-SNOM in the citations of this publication. Nevertheless, at the same time it is important to state that the Keilmann and Hillenbrand group are the inventors of the method and several specific measurement modes such as Ps-Het and nano-FTIR spectroscopy were developed by them. Therefore, it is reasonable to cite their work for explaining the specific measurement modes and the membrane based liquid s-SNOM method. Furthermore, we have already cited in the manuscript several works of other scientist conducting s-SNOM studies such as:

Cernescu, A. et al. Label-Free Infrared Spectroscopy and Imaging of Single Phospholipid Bilayers with Nanoscale Resolution. *Analytical chemistry* 90, 10179–10186; 10.1021/acs.analchem.8b00485 (2018).

Möslein, A. F., Gutiérrez, M., Cohen, B. & Tan, J.-C. Near-Field Infrared Nanospectroscopy Reveals Guest Confinement in Metal-Organic Framework Single Crystals. *Nano letters* 20, 7446–7454; 10.1021/acs.nanolett.0c02839 (2020).

Hauer, B., Engelhardt, A. P. & Taubner, T. Quasi-analytical model for scattering infrared near-field microscopy on layered systems. *Optics express* 20, 13173–13188; 10.1364/OE.20.013173 (2012).

Chen, X. et al. Modern Scattering-Type Scanning Near-Field Optical Microscopy for Advanced Material Research. *Advanced materials (Deerfield Beach, Fla.)* 31, e1804774; 10.1002/adma.201804774 (2019).

Actions taken:

We now additionally cite the following works to better reflect the diversity of the scientific community in the near-field microscopy field:

- Hu, H. et al. Gate-tunable negative refraction of mid-infrared polaritons. *Science* (New York, N.Y.) 379, 558–561; 10.1126/science.adf1251 (2023).
- Eliaz, D. et al. Micro and nano-scale compartments guide the structural transition of silk protein monomers into silk fibers. *Nature communications* 13, 7856; 10.1038/s41467-022-35505-w (2022).
- Nishida, J. et al. Sub-Tip-Radius Near-Field Interactions in Nano-FTIR Vibrational Spectroscopy on Single Proteins. *Nano letters* 24, 836–843; 10.1021/acs.nanolett.3c03479 (2024).

Furthermore, we added the following section in the discussion of the manuscript on **page 16** to take account of the different methods and publications in the field of liquid s-SNOM citing several papers of groups conducting liquid s-SNOM:

"There are important performance differences between our membrane based liquid s-SNOM method and competing methods, which include in-liquid s-SNOM. In general, approaches that measure s-SNOM or AFM-IR directly in liquid can provide spatial resolutions only limited by the tip and recording of the full topography^{50–54}, whereas membrane-based method is limited to around 100 nm. However, these approaches all employ an ATR-based transmission illumination, which is very difficult to align. Moreover, our method enables more robust and longer duration experiments, because the tip is protected from contamination by the sample and the optical alignment is maintained, allowing more complex dynamic studies of, e.g., living cells¹⁸. Thinner membranes could be utilized to further increase the spatial resolution limit of our method, as long as they provide sufficient mechanical stability for water adhesion and AFM tapping. For example, liquid s-SNOM studies have been conducted with ultra-thin graphene capping layers^{55–57} and metal oxide capping layers such as TiO₂ and Al₂O₃⁵⁸, which enable higher resolution imaging in liquid. Building on this idea, it has been demonstrated that a liquid sample containing viruses can be wrapped in a graphene sheet and investigated with s-SNOM enabling true nanoscale imaging and topography recording with a capping layer⁵⁹. These capping layers have the additional benefit of having no phonon resonance in the MIR compared to SiN, which exhibits a strong resonance in the range of 800 to 1150 cm⁻¹ masking important molecular infrared resonances¹⁸. From a technological perspective, the wide and cheap commercial availability of the SiN membranes enables straightforward consecutive experiments on complex biological systems, as opposed to other capping layers, which need to be self-fabricated for every experiment."

In the paragraph we cited the following paper from worldwide s-SNOM scientists:

- Virmani, D. et al. Amplitude- and Phase-Resolved Infrared Nanoimaging and Nanospectroscopy of Polaritons in a Liquid Environment. *Nano letters* 21, 1360–1367; 10.1021/acs.nanolett.0c04108 (2021).
- O'Callahan, B. T. et al. In Liquid Infrared Scattering Scanning Near-Field Optical Microscopy for Chemical and Biological Nanoimaging. *Nano letters* 20, 4497–4504; 10.1021/acs.nanolett.0c01291 (2020).
- Wang, H. et al. Liquid-Phase Peak Force Infrared Microscopy for Chemical Nanoimaging and Spectroscopy. *Analytical chemistry* 93, 3567–3575; 10.1021/acs.analchem.0c05075 (2021).
- Jin, M., Lu, F. & Belkin, M. A. High-sensitivity infrared vibrational nanospectroscopy in water. *Light, science & applications* 6, e17096; 10.1038/lsa.2017.96 (2017).
- Pfitzner, E. & Heberle, J. Infrared Scattering-Type Scanning Near-Field Optical Microscopy of Biomembranes in Water. *The journal of physical chemistry letters* 11, 8183–8188; 10.1021/acs.jpcclett.0c01769 (2020).
- Lu, Y.-H. et al. Infrared Nanospectroscopy at the Graphene-Electrolyte Interface. *Nano letters* 19, 5388–5393; 10.1021/acs.nanolett.9b01897 (2019).
- Meireles, L. M. et al. Synchrotron infrared nanospectroscopy on a graphene chip. *Lab on a chip* 19, 3678–3684; 10.1039/C9LC00686A (2019).
- Zhao, X. et al. In vitro investigation of protein assembly by combined microscopy and infrared spectroscopy at the nanometer scale. *Proceedings of the National Academy of Sciences of the United States of America* 119, e2200019119; 10.1073/pnas.2200019119 (2022).
- Lu, Y.-H. et al. Ultrathin Free-Standing Oxide Membranes for Electron and Photon Spectroscopy Studies of Solid-Gas and Solid-Liquid Interfaces. *Nano letters* 20, 6364–6371; 10.1021/acs.nanolett.0c01801 (2020).

Reviewer #2 (Remarks to the Author):

The manuscript entitled "Transient infrared nanoscopy resolves the millisecond photoswitching dynamics of single lipid vesicles in water" offers an experimental investigation into the resolution of lipid vesicle photoswitching dynamics, which is relevant to advancements in nanomedicine, photopharmacology, and drug delivery systems. The authors employ transient infrared nanoscopy, clearly describing the method and its ability to achieve the spatio-temporal resolution (150 nm-30 ms), specifically focusing on photoswitchable liposomes within a liquid cell. This methodological approach significantly contributes to the practical application of nanoscopy in liquid experiments. Overall, the study is well-conducted, presenting a novel approach to examining lipid vesicle dynamics through transient infrared nanoscopy. The article demonstrates a degree of innovation and is likely to attract interest from researchers within this domain. I recommend its publication in Nature Communications following several minor revisions.

Our response:

We thank the reviewer for their recognition of the quality of our study. We hope that the comments and changes to the manuscript will clarify the below stated points.

Reviewer comment:

1. In Figures 2 and S3, regarding the flattening of the lipid nanovesicles, could the authors provide the complete topography image rather than only the line cut? I am particularly interested in the overall shape of the flattened lipid nanovesicle on the SiN membrane.

Our response:

We agree with the reviewer that the complete topographic image should be provided to better understand the overall behavior of the vesicles on the membrane.

Actions taken:

We added the topography image of the overview scan corresponding to **Figure S3** with the boxes highlighting the different vesicles analyzed in **Figure S3** with the following caption.

Figure S3: (a) Topographic overview image recorded in correlation to the near-field optical amplitude and phase shown in Fig. 2a and b. The white boxes highlight the positions of the particles studied with the line cuts in b – e, scale bar 5 μm .

Reviewer comment:

2. The abstract and section title claim a temporal resolution of 50 ms, as stated in the phrase "50 ms resolved MIR near-field signal tracking of the photoswitching dynamics of a single lipid vesicle." However, the main text and Figure S8 present results at 30 ms, which show a 3.6 sigma, exceeding the temporal resolution limit. This discrepancy should be addressed.

Our response:

We thank the reviewer for the careful reading of the abstract and the manuscript. Indeed, our maximum temporal resolution is 30 ms. We apologize for the mistake, which was introduced when compiling the final version of the manuscript for submission and have now corrected the value throughout.

Actions taken:

We changed the maximal time resolution throughout the manuscript to 30 ms.

Reviewer comment:

3. As the lateral resolution of 150 nm - 200 nm is relatively large for typical nanoscopy in liquid ["Amplitude- and Phase-Resolved Infrared Nanoimaging and Nanospectroscopy of Polaritons in a Liquid Environment", Nano Lett. 2021, 21, 1360–1367; "In Liquid Infrared Scattering Scanning Near-Field Optical Microscopy for Chemical and Biological Nanoimaging", Nano Lett. 2020, 20, 4497–4504; "Liquid-Phase Peak Force Infrared Microscopy for Chemical Nanoimaging and Spectroscopy", Analytical Chemistry 2021 93 (7), 3567-3575; "High-sensitivity infrared vibrational nanospectroscopy in water", Light: Science & Applications (2017) 6, e17096; "Antenna array-enhanced attenuated total reflection IR analysis in an aqueous solution", Nanoscale, 2019,11, 18543-18549], which claim typical spatial resolution of a few tens of nm. Moreover, the liquid cell does not allow for the observation of the complete shape of the lipid nanovesicle. The authors should discuss potential methods for overcoming these limitations.

Our response:

We thank the reviewer for raising this point. We agree that the resolution of our approach compared the other mentioned s-SNOM methods should be discussed the manuscript. Furthermore, we agree

that a discussion of solutions to increase the resolution such as using thinner membranes or wrapping the liquid sample in graphene should be included.

Actions taken:

We added the following paragraph in the discussion section of the manuscript on **page 16** to set our method in the context of existing literature on liquid s-SNOM. In this discussion we also cited several of the above-mentioned papers as we believe they are important for the readers. This discussion lists several points to increase the spatial resolution and enable whole sample measurements in liquid.

"There are important performance differences between our membrane based liquid s-SNOM method and competing methods, which include in-liquid s-SNOM. In general, approaches that measure s-SNOM or AFM-IR directly in liquid can provide spatial resolutions only limited by the tip and recording of the full topography⁵⁰⁻⁵⁴, whereas membrane-based method is limited to around 100 nm. However, these approaches all employ an ATR-based transmission illumination, which is very difficult to align. Moreover, our method enables more robust and longer duration experiments, because the tip is protected from contamination by the sample and the optical alignment is maintained, allowing more complex dynamic studies of, e.g., living cells¹⁸. Thinner membranes could be utilized to further increase the spatial resolution limit of our method, as long as they provide sufficient mechanical stability for water adhesion and AFM tapping. For example, liquid s-SNOM studies have been conducted with ultra-thin graphene capping layers⁵⁵⁻⁵⁷ and metal oxide capping layers such as TiO₂ and Al₂O₃⁵⁸, which enable higher resolution imaging in liquid. Building on this idea, it has been demonstrated that a liquid sample containing viruses can be wrapped in a graphene sheet and investigated with s-SNOM enabling true nanoscale imaging and topography recording with a capping layer⁵⁹. These capping layers have the additional benefit of having no phonon resonance in the MIR compared to SiN, which exhibits a strong resonance in the range of 800 to 1150 cm⁻¹ masking important molecular infrared resonances¹⁸. From a technological perspective, the wide and cheap commercial availability of the SiN membranes enables straightforward consecutive experiments on complex biological systems, as opposed to other capping layers, which need to be self-fabricated for every experiment."

Reviewer #3 (Remarks to the Author):

Gölz et al. demonstrated a novel transient MIR nanoscopy method for investigating photoswitching dynamics in situ on nanocarriers and showcase this approach on individual photo-switchable lipid nanocarriers (lipid vesicles) isomerisation at the single liposome level in aqueous environments. Golz and co-authors present the first s-SNOM in situ study on actively photo-induced nanocarrier structure-property relationship change modulated by ultraviolet and blue light illumination at a temporal resolution tolerant for practical applications (50 ms, can be considered as being real time in many cases).

This manuscript is well-written and easy to follow in a clear structure based on well-thought experimental design and convincing s-SNOM experimental results. The results have demonstrated the unique advantage of MIR s-SNOM over typical ATR-FTIR and far-field spectroscopy approaches. This remarkable study is a step changer for near-field optics and nanophotonics based the group's solid experimental studies published before. The demonstrated approach in this manuscript is a milestone towards studying other unlabelled soft matter complex dynamics under native physiological conditions via light-matter interactions in the future.

The nanoscopy approach in this manuscript will be a new research paradigm for revealing the microscopic origin on fundamental biological and chemical processes in native liquid environment by utilising nanoscale light-matter interactions, offering unparalleled insights for material/chemical scientists as well as microbiologists in future studies beyond lipid systems.

I recommend a minor revision, and the authors are expected to address my comments below before publishing on Nature Communications.

Our response:

We thank the reviewer for the kind comments on our manuscript and hope we can resolve the mentioned concerns.

Reviewer comment:

1. [Relevant to the liquid cell] The vesicles are attached at the bottom surface of SiN membrane via van der Waals (vdW) forces, which may be considered as not as native as free running molecules in liquid. In some certain chemical reactions, this externally introduced vdW forces may modulate microscopic biological or chemical process (since the vesicles may be flattened or deformed when adhering to the SiN membrane). Also, the SiN may be reactive to attached biomolecules or chemical compound if we want to study at the single molecular level. Are we still probing deformed molecules, which is not as native, in liquid environment? It will be helpful to clarify for non-s-SNOM experts: (1) is there a strategy to mitigate this issue or (2) do we need to find application scenarios to study

chemical process not perturbed by nanoscale adhesion (or study problems directly relevant to nano-adhesion)?

Will other 10-nm thin film other than SiN be able to serve as capping layer for control experiments to rule out potential artefacts introduced by the (SiN) membrane or by vdW forces introduced by capping layers itself?

What kind of requirements (e.g., dielectric and mechanical properties) of the capping layer required for this s-SNOM approach? For the wavelength range outside of nano-FTIR, is there other candidates of capping layers? For example, being transparent (dispersion-less) to both pumping & probe radiation wavelength but sensitive to the interrogated sample solely?

Our response:

We thank the reviewer for being interested in the liquid s-SNOM method. Ultimately, the liquid s-SNOM method is a scanning probe method and all scanning probe methods rely on samples that are attached/ adhere to some kind of surface or supporting structure. Still these methods are widely employed in life science and chemistry. Therefore, we do not think that the adhesion is a critical showstopper for studying samples in-situ. Furthermore, the SiN membrane technique is already well established in the scientific community of in-situ TEM, SEM and X-ray spectroscopy and imaging providing valuable new insights in many fields.

Considering the question of chemical reactivity of the SiN membrane towards the sample. The SiN membrane is a chemically very inert material and will thus not react with samples under normal conditions (standard pH-level, non-oxidative or reductive samples, moderate temperatures). Therefore, we believe that the chemical influence on the sample can be neglected as it behaves more like a substrate layer.

Additionally, one could circumvent the adhesion problem by experimental design depending on the sample systems to be studied. As stated in more detail in response to comment 5 one could for example use a supported bilayer to study vesicle fusion and drug release circumventing the problem of vesicle adhesion.

The reviewer is correct with his intuition that other capping layer materials might be beneficial. There are several publications (listed below) that use capping layers such as graphene or metal oxides with the advantage of them being thinner and having no optical phonon in the MIR range. However, we want to highlight that the 10 nm SiN membrane is commercially available and has no need for

extensive nanofabrication compared to the metal oxide or 2D material membranes, which are not commercially available. This fact leads also to a higher reproducibility of the experiments:

Lu, Y.-H. et al. Infrared Nanospectroscopy at the Graphene-Electrolyte Interface. *Nano letters* 19, 5388–5393; 10.1021/acs.nanolett.9b01897 (2019).

Meireles, L. M. et al. Synchrotron infrared nanospectroscopy on a graphene chip. *Lab on a chip* 19, 3678–3684; 10.1039/C9LC00686A (2019).

Zhao, X. et al. In vitro investigation of protein assembly by combined microscopy and infrared spectroscopy at the nanometer scale. *Proceedings of the National Academy of Sciences of the United States of America* 119, e2200019119; 10.1073/pnas.2200019119 (2022).

Lu, Y.-H. et al. Ultrathin Free-Standing Oxide Membranes for Electron and Photon Spectroscopy Studies of Solid-Gas and Solid-Liquid Interfaces. *Nano letters* 20, 6364–6371; 10.1021/acs.nanolett.0c01801 (2020).

Khatib, O. et al. Graphene-Based Platform for Infrared Near-Field Nanospectroscopy of Water and Biological Materials in an Aqueous Environment. *ACS nano* 9, 7968–7975; 10.1021/acsnano.5b01184 (2015).

There are several requirements in our opinion that needs to be satisfied by the capping layer for liquid s-SNOM operation. Firstly, the membrane needs to be thin enough for the near field to penetrate into the liquid sample. Secondly, the membrane needs to be mechanically robust enough for the AFM of the s-SNOM to be operated in tapping mode with water adhere underneath the membrane. Next, the membrane should be as transparent as possible possible to the wavelength utilized for the near-field investigation of the sample. This means a low refractive index and very small absorption losses in the desired wavelength. Finally, we want to emphasize the above point again. For practical experiments it is important to have relative cheap membranes with reproducible quality and good availability to optimize the liquid experiment with many measurements.

Actions taken:

We added the following paragraph to the discussion section of the manuscript on **page 16** to introduce different liquid s-SNOM concepts and strategies (different capping layers, in liquid s-SNOM) to the readers of the manuscript with their inherent advantages and disadvantages:

"There are important performance differences between our membrane based liquid s-SNOM method and competing methods, which include in-liquid s-SNOM. In general, approaches that measure s-SNOM or AFM-IR directly in liquid can provide spatial resolutions only limited by the tip and recording of the full topography^{50–54}, whereas membrane-based method is limited to around 100 nm. However, these approaches all employ an ATR-based transmission illumination, which is very difficult to align. Moreover, our method enables more robust and longer duration experiments, because the tip is protected from contamination by the sample and the optical alignment is maintained, allowing more complex dynamic studies of, e.g., living cells¹⁸. Thinner membranes could be utilized to further increase the spatial resolution limit of our method, as long as they provide sufficient mechanical stability for water adhesion and AFM tapping. For example, liquid s-SNOM studies have been conducted with ultra-thin graphene capping layers^{55–57} and metal oxide capping layers such as TiO₂ and Al₂O₃⁵⁸, which enable higher resolution imaging in liquid. Building on this idea, it has been demonstrated that a liquid sample containing viruses can be wrapped in a graphene sheet and investigated with s-SNOM enabling true nanoscale imaging and topography recording with a capping

layer⁵⁹. These capping layers have the additional benefit of having no phonon resonance in the MIR compared to SiN, which exhibits a strong resonance in the range of 800 to 1150 cm⁻¹ masking important molecular infrared resonances¹⁸. From a technological perspective, the wide and cheap commercial availability of the SiN membranes enables straightforward consecutive experiments on complex biological systems, as opposed to other capping layers, which need to be self-fabricated for every experiment."

Reviewer comment:

2. The authors deliberately mis-align two LEDs (365 nm & 465 nm) off the tip to avoid creating near-fields at non-IR wavelengths. If this two LEDs were aligned at the tip simultaneously, will the reported results of photo-switchable behaviour / nano-FTIR spectral results change? The authors could consider adding these results as control experimental results. This could create motivation in s-SNOM community for another multi-colour s-SNOM modality spanning from THz to UV for transient IR nanoscopy or even transient THz nanoscopy for using this kind of liquid cell.

Our response:

We thank the reviewer for the suggestion of conducting visible and infrared nanoscopy at the same time in analogy to the recently published multi-colour s-SNOM modality. In our opinion it could be in principle possible to align both the visible light beam from the left and the MIR-beam from the right side of the parabolic mirror onto the same tip and collect the scattering signal. However, this is an extremely difficult and time-consuming experiment, which is way beyond the scope of this paper. In addition, to perform simultaneous VIS and MIR near-field measurements the NeaSNOM microscopes needs two NeaDAQ lock-in amplifier cards to demodulated both the VIS and MIR detector signal. Our microscope does not have these kinds of electronics and therefore we cannot perform the measurements. Finally, we want to emphasize that we deliberately misalignment the beam to cover the whole membrane area with a size of 250 μm x 250 μm to switch all the vesicles below the membrane and not only a specific part. We like to refer the reviewer to **Fig. S9** and **Fig. S10**, which show reference measurements of the s-SNOM signal on D₂O and Si, when varying the illumination wavelength of the LED. There is no change in detector signal due to change in wavelength. Thus, we believe that the UV-Vis light source does not induce an artefact in the transient time trace measurements.

Action taken:

We added the following sentence to the discussion on **page 17**, mentioning the perspective of multi-color s-SNOM modalities for biospectroscopy:

"Another interesting avenue to explore would be the application of multi-color s-SNOM imaging to simultaneously record the response at several wavelengths of the sample in liquid undergoing dynamic processes such as neurotoxic protein aggregation⁴⁵ "

Reviewer comment:

3. The authors showed 1-nm membrane protrusion (mechanical footprint) with pronounced optical signals due to vesicles from nano-FTIR confirmation. For non-trivial topography scanned by s-SNOM, is there a way to remove potential spurious optical signal contribution not relevant to the interrogated sample itself, especially when multiple high harmonics are not possible? This could be crucial when we heat up the liquid cell add solvents (control PH) to trigger other biochemical processes.

Discussion on these will be helpful for readers when future researchers adopting this liquid cell for experimental design or following this method to prepare customised membranes / liquid cells, which does not have reproducible flatness as a commercial membrane.

Our response:

There are multiple ways to treat this potential limitation. First, optimizing fabrication processes to achieve surfaces that are as flat as possible is crucial for the success of our method, since it not only increases the stability of the tapping tip, but will likely facilitate the attachment of objects to the underside of thin membranes. Since the interactions between membrane and sample mainly occurs through Van-der-Waals forces, surface roughness could be a significant limitation to the wettability and deposition of the sample on the membrane surface.

That being said, in principle, using higher harmonics would not substantially help in this case, since the surface deformations would still potentially be directly underneath the tip and thus a researcher using our method would experience those same artefacts. A potential solution would be to use image reconstruction algorithms in order to identify potential defects in the membrane surface via the topographical image measured simultaneously with the optical images, then correlate both images and replace the affected pixels (something similar has been done in the following paper using s-SNOM to generate surface phonon polaritons in thin membranes <https://doi.org/10.1038/s41566-024-01410-5>. There is a section dedicated to this problem in the SI of the publication.).

Another way to prevent local deformations, which can be caused by the tapping tip, would be to increase or decrease the tapping amplitude to minimize membrane deformation. We show this in the following paper <https://doi.org/10.1038/s41566-024-01410-5>, where we used the same SiN membranes as used in this work. The optimal tapping amplitudes for our specific system were between 60 and 80 nm.

Actions taken:

We added the following sentence to the Method Section on **page 17**:

"In principle, choosing tapping amplitudes between 60 and 80 nm reduces deformation of the membrane by the tapping tip and is a good compromise between SNR and a background free signal³¹."

We also added the following section to the Discussion on **page 15**:

" A potential solution to removing unwanted topographical artefacts caused by surface roughness is to use image reconstruction algorithms to correct optical images via the simultaneously measured topography⁴⁵, as well as improving fabrication methods."

Reviewer comment:

4. For MIR optical phase raw images collected by the tuneable monochromatic laser (e.g., Fig. 3b, Fig. 4b/f, or Fig. S4), I can observe non-negligible non-zero phase signals at pixels not overlapping with optical amplitude (which we consider they are correlating to sample flattened topography attached

below the membrane). Is there other interpretation other than background noise on them (e.g., Fig.4b and Fig. S4 4th column counting from the left to right hand side)?

Our response:

The non-negligible non-zero phase signal is (in our opinion and experience) a mixture of different effects. Firstly, all these measurements have been conducted on the photoswitching resonance located at around 1606 cm^{-1} . This resonance is lower in intensity in comparison to the carbonyl resonance and therefore the SNR of the phase images are much lower. The low SNR results in a higher relative background signal and the observed non-zero background phase signal of the liquid. In addition, some of the strip like phase signals on the water can be assigned to crosstalk of the AFM signal into the optical near-field signal as the AFM operation is not always as stable as when operated on a flat piece of silicon. Finally, some of the phase signal can might also originated from some lipid contamination attached below the membrane.

Reviewer comment:

5. Since we cannot measurement directly the sample topography in this exciting liquid cell achievement, if researchers would like to track photo-release bio-chemical process, e.g., drug delivery, will the current approach (or revised approach potentially) be able to track this free running process (not attached to the membrane) by tuning the probing depth with tip tapping amplitude potentially? How shall we be confident what are we measuring below (but not attached to) the membrane bottom surface?

Our response:

We thank the reviewer for bringing up this point. In principle, tracking the phototriggered release of bioactive molecules from a lipid nanoparticle should also be possible with a vesicle that adheres to the membrane. Our photoswitching study has shown that the vesicle can still dynamically and reversibly change its shape. Therefore, we are confident that a drug could still be released from a system through bilayer lipid membrane opening. Hereby, nano-FTIR can first identify the chemical fingerprint of the drugs in the vesicle. We have shown in an earlier study (Kaltenecker, K.J., et al. *Sci Rep* 11, 21860 (2021)) that we can probe more than 100 nm into the liquid sample compartment, when using the second demodulation order. A probing depth deep enough to probe into the cargo section of lipid nanoparticles, which are only 100 to 200 nanometers large. Subsequently, the release could be tracked in time by our demonstrated signal tracing method or by a second nano-FTIR spectrum that confirms the release of the drug molecules.

In addition, one could think of a system being composed of a supported bilayer membrane simulating a cell membrane and lipid nanoparticles below that get phototriggered and fuse with the supported membrane. Again, the more than 100 nm probing depth should be enough to probe through the supported bilayer and characterize the cargo of the lipid nanoparticle. In this system the vesicle would

not adhere to the SiN membrane, and it would mimic a natural uptake of a lipid nanoparticle drug system.

Actions taken:

We added the following sentence to the discussion section of the manuscript on **page 15** to give the readers an idea of potential ways to circumvent the adhesion problem through a more sophisticated experimental design:

"Here, one could consider first forming SPBs on the SiN membrane and then studying the fusion of liposomes with the lipid bilayer, mimicking a cellular environment and avoiding the problem of the vesicles adhering on the SiN membrane, which may influence isomerization dynamics."

Reviewer comment:

6. How will the tip approach curve look like at the spatial position with vesicles and with the sucrose solution (D₂O)? Can the authors therefore use finite dipole model to predict the probing depth for the liquid cell (considering the 10-nm Si membrane)

Our response:

The probing tip can be marginally increased by increasing the tapping amplitude, as is well known from the s-SNOM literature. However, the biggest factor in determining probing depth is the tip radius r . In general, the tip probing depth is approximately proportional to r . In an earlier publication (<https://doi.org/10.1038/s41598-021-01425-w>), we found that for a tip of size $r=60$ nm (which are the same tips used in this study), the probing depth is approx. 100 nm (where the amplitude and phase signals drop to $1/e$ of the maximum). When considering the finite dipole model (FDM), it is in general less useful for complex biological samples if the exact dielectric function is unknown. However, it is certainly possible to use the FDM to consider a three-layer system with 10 nm SiN as the first layer, D₂O as a variable second layer and a material with a known dielectric function (such as PMMA) as an infinitely thick substrate. We have done this calculation as well in (<https://doi.org/10.1038/s41598-021-01425-w>), Fig. S3, only with H₂O instead of D₂O.

To find the approximate probing depth for our system, we can consider a three-layer-system (10 nm SiN, D₂O of variable thickness and PMMA as substrate). Below are calculated spectra for different thicknesses of D₂O between 0 and 200 nm at two different tip radii, $r=20$ nm and $r=60$ nm:

As expected, the resonance at 1740 associated with PMMA decreases with larger D₂O thicknesses, while the resonance of D₂O at around 1200 increases. Additionally, larger tip radii in general yield higher probing depths. In this way, the probing depth of the tip used for the experiment can be estimated.

Actions taken:

We added the following sentence to the discussion on **page 16** to highlight that the probing depth is highly tip radius dependent:

"This would also reduce the near-field probing depth to 30 nm as the near-field probing depth is highly dependent on the tip¹⁸."

Reviewer comment:

7. Have the authors explored the largest tapping amplitude suitable for this kind of in situ s-SNOM measurements using liquid cell? For certain wavelength like THz nanoscopy, the usually adopted tapping amplitude could be > 200 nm or 300 nm for measurable tip-scattered signals. Will a 100-nm SiN membrane be mechanically strong enough? This kind of information will be useful for translating this approach to studies in applications for a broad readership.

Our response:

We have shown in the following publication (<https://doi.org/10.1002/sml.202402568>) through repeatable measurements that the AFM can be operated at tapping amplitudes of up to 200 nm on a SiN membrane without destroying the membrane. Of course, such tapping amplitudes would be too high for the wavelength range we operate in, since it would mostly result in adding unnecessary background. However, for THz nanoscopy, our approach would be suitable, since the SiN membranes we used in our study seem to hold up even against comparatively large loading forces. We have even observed in experiments not shown here that tapping amplitudes up to 400 nm are possible. These large tapping amplitudes will cause larger deformations in the membrane and might cause mechanical

stress on the sample, such high tapping amplitudes do not break or damage the membrane in any significant way.

Actions taken:

We have added the following sentence in the discussion on **page 17**:

"Additionally, the method is compatible with correlated measurements such as fluorescence and Raman imaging, as well as for THz nanoscopy, for which higher tapping amplitudes (>200 nm) are mechanically possible."

Reviewer comment:

8. I noticed the g parameter in finite dipole model is 0.6 as a real number, will the used parameters be able to recover the measured approach curve at the monochromatic frequency?

Our response:

We thank the reviewer for pointing out that we wrote the value of the g parameter as 0.6. This was a mistake on our part, since we actually used the more accurate value of $g=0.7 \cdot e^{i0.06}$ (as shown in <https://opg.optica.org/oe/fulltext.cfm?uri=oe-15-14-8550&id=138960>) for our modeling.

Actions taken:

We corrected the mistake in the Method section on **page 20**:

"Results are shown in Fig. S3, using the modeling parameters $a = 80$ nm (tapping amplitude), $r = 60$ nm (tip radius), $L = 300$ nm (length of spheroid), $g = 0.6 + 0.7e^{i0.06}$ (charge induced in the tip)."

Reviewer comment:

9. In many nano-FTIR literature, researchers may use the imaginary part of normalised s-SNOM signal to locate the spectral absorption peaks. The authors chose to use phase signals directly in this manuscript for assignment of functional group, any specific reason? A more reasonable spectral peak position for functional group assignment?

Our response:

We thank the reviewer for bringing up this point. We agree that the imaginary part calculated via $\text{Im}[\sigma_n(\omega)] = s_n(\omega)\sin[\varphi_n(\omega)]$ can be used for plotting the nanoscale absorption spectrum of the sample. However, plotting the optical phase spectrum is a very common practice in the scientific s-SNOM literature as it directly correlates to the IR absorption. In addition, we believe that the near-field phase is easier to interpret compared to the imaginary part of the normalized s-SNOM signal because it shows a linear increase in phase signal with increasing sample material. In contrast, the imaginary part does not linearly increase with the sample thickness and the peak position can shift significantly as it contains a mix of sample and substrate information due to the optical amplitude contribution.

Therefore, we opted for the nano-FTIR phase for the photoswitching spectra. The following publication explains the above stated relation in more detail:

Mastel, S., Govyadinov, A. A., Oliveira, T. V. A. G. de, Amenabar, I. & Hillenbrand, R. Nanoscale-resolved chemical identification of thin organic films using infrared near-field spectroscopy and standard Fourier transform infrared references. *Applied Physics Letters* 106; 10.1063/1.4905507 (2015).

Actions taken:

We added the following sentence to the method section of the manuscript on **page 19** to better explain for the reader the use of the optical phase and not the imaginary part of the near-field signal and we cited the above-mentioned publication:

"We choose to plot the optical phase ϕ_n as opposed to the sometimes utilized imaginary part of the near-field signal as it correlates linearly with the infrared absorption of the sample in contrast to the imaginary part⁶⁶."

Reviewer comment:

10. Since the resonance peak intensity decrease at 1606 cm⁻¹ is the indication of trans-to-cis isomerisation after 365-nm illumination, it is better to highlight how long the whole measurement took (from topping up liquid nitrogen to finish the nanospectroscopy) as future benchmark. The MIR phase and amplitude intensity decrease could be perceived mistakenly as correlating to MCT detector cooling performance.

Our response:

We thank the reviewer for being interested in the duration of the measurements. The two spectra showing the different photoswitching states of the vesicle have been recorded consecutively. One spectrum is an average of 20 single nano-FTIR spectra and has a total acquisition time of 14 min. The workflow for characterizing the photoswitchable lipid vesicle was as following: We aligned the s-SNOM optics in reflection mode, recorded a reference nano-FTIR spectrum for normalization on a clean piece of silicon and prepared the liquid lipid sample suspension. Next, we conducted a white-light overview scan to identify several lipid vesicles in the liquid sample. Subsequently, we record a close-up scan of a single lipid vesicle like the image shown in **Fig. 1e**. Finally, the nano-FTIR spectra have been recorded on top of the single vesicle with blue and UV illumination to characterize the *trans*- and *cis*-state of the single vesicle. In general, the whole workflow takes about 2 to 3 h. In our experience the MCT detector has a stable signal over 8 h. Also, if the cooling of the liquid nitrogen is not sufficient anymore no signal at all can be detected. Therefore, the differences in intensity cannot originate from the cooling performance of the detector.

In addition, we would like to highlight the fact that the phase signal of the carbonyl peak at 1742 cm⁻¹ is of the same value in the nano-FTIR spectrum of the *trans*- and *cis*-state shown below. This is expected from the far-field ATR-FTIR spectrum shown in below and in **Fig. S1**. If there would be a drift in the detector or laser intensity the carbonyl intensity would not be the same value anymore.

Therefore, we infer that the change in intensity at the photoswitching peaks is a result of the molecular switching of the molecules and cannot originate from a signal drift.

ATR-FTIR reference spectra:

Nano-FTIR spectra:

Actions taken:

We added the following sentence to the results part of the manuscript on **page 5**. Highlighting that the constant signal of the carbonyl peak is a proof for a stable spectroscopic system and that the change in intensity at the other resonances are due to the molecular photoswitching:

"Importantly, the carbonyl signal at 1742 cm⁻¹ in the nano-FTIR spectrum under 365 nm illumination does not change in intensity compared to 465 nm illumination, as is expected from the reference ATR-FTIR spectrum. This confirms that there was no change in laser power, optical tip alignment, or detector responsivity between the recording of the two spectra, which could lead to intensity modulations."

We further agree that it is important to state the time difference between the two spectra shown in **Fig. 1f** and added the following sentence to the method section of the manuscript on **page 19**:

"The nano-FTIR spectra were recorded with an interferometer scan length of 300 μm , resulting in a nominal spectral resolution of 14.2 cm⁻¹. During the interferometer movement, 1200 discrete data points were recorded with an integration time of 35 ms per point. The shown spectra are an average of 20 nano-FTIR spectra recorded consecutively resulting in a total acquisition time of 14 min for one spectrum shown in **Fig. 1f**. Importantly, both averaged spectra in **Fig. 1f** have been recorded consecutively to minimize the time difference between the spectra resulting in unwanted outside influence on the spectroscopic response of the sample. The nano-FTIR spectra were recorded with such a relatively low spectral resolution as the SNR of a FTIR spectrum and the spectral resolution correlate inversely⁹ and for liquid s-SNOM measurement on a liquid sample fast measurement with high SNR is essential."

Reviewer comment:

11. The author used a wire grating to attenuate the output power (tunable laser) to about 3 mW before reaching the beam splitter, what is the reason?

Our response:

We thank the reviewer for being interest in the setup of the study. In our experience the best SNR for Ps-Het measurement can be achieved by using a laser power of 3 to 5 mW before reaching the

beamsplitter. The attenuated laser power is also low enough to avoid sample and tip damage. This is also mentioned in the following publication:

Vicentini, E. *et al.* Pseudoheterodyne interferometry for multicolor near-field imaging. *Optics express* **31**, 22308–22322; 10.1364/OE.492213 (2023).

As the un attenuated output power of the laser is higher we must attenuate the beam with the grid attenuator.

Actions taken:

We clarify the reason for using the grid attenuator by adding the following sentence to the method section of the manuscript on **page 18**:

"The output power of the laser is attenuated by a wire grating (Lasnix, Berg, Germany) to about 3 to 5 mW before reaching the beamsplitter, which is commonly understood to be the optimal range for achieving high SNR and low enough for avoiding tip and sample damage in Ps-Het measurement mode."

Reviewer comment:

12. [p4, line 114] It is better to describe trans & cis for non-Latin readers without chemical background. Fig. 1d clarifies cis-trans isomerism [p4, line 113] visually --- consider adding one sentence to describe these two in the text.

Our response:

We thank the reviewer for point out this point. It makes sense to further explain the *trans-cis* isomerization in more detail.

Actions taken:

We added the following sentence in brackets to the main section of the manuscript on **page 4** to further explain the specific kind of isomerization:

"Illumination of the lipid vesicles with a wavelength of 365 nm triggers the isomerization of the azobenzene moiety from *trans* to *cis* (resulting in a change of the bulky phenyl rings from being located

opposite of each other to besides each other and drastically increasing the spatial footprint of the tail group of the lipid), while, ϵ ."

Reviewer comment:

13. Fig. S2a is blurry. It is hard to see the functional group in purple circle, considering the vector figure. Minor: the red colour for carbon chain does not match with the colour in panel b & c (Fig. S2).

Our response:

We thank the reviewer for the careful reading of the manuscript. Indeed, **Fig S2a** is blurry, and the color is slightly off.

Actions taken:

We inserted the following higher resolution **Figure S2** with the correct colour to match panel b & c in the supporting information on **page 2**.

Reviewer #4 (Remarks to the Author):

The manuscript reports experimental approach seeking to probe the morphology and dynamics of a phospholipid vesicle containing photoswitchable azo-lipids. They performed in-situ s-SNOM studies on actively switched vesicles, by repeatedly and reproducibly switching between cis and trans states using UV/blue lights, with a spatio-temporal resolution of 100-150 nm and 50 ms. These studies are useful for understanding the photo-induced drug release processes at the cellular level. s-SNOM is a powerful and label-free surface sensitive technique for surface characterizations. The technique involves an AFM tip oscillating vertically above the sample surface and a focused pulsed/CW laser scattering from the apex of the tip. The scattered beam of light contains information about near-field interactions between the sample and the tip, thereby leading to a high spatial resolution. They performed in-situ photo-induced dynamic measurements on single lipid vesicle level.

Our response:

We thank the reviewer for their comments on our manuscript and hope we can solve the mentioned concerns.

Reviewer comment:

1. Although the results obtained are partially insightful, the manuscript fails to substantiate the primary claims as I have described below. In the abstract, the authors claim that their study leverages chemical recognition which I agree only partially. I do not see a significant difference in the spectrum shown in Figure S1 between the cis and trans states of the azo-PC. Here, the authors rely solely on the change in intensity. However, their previous measurements (Crea et al., Front. Mol. Biosci., 9, 905306, 2022) demonstrated that the trans states show breathing modes at 1603 and 1580 cm^{-1} whereas the cis states show at 1511 and 1496 cm^{-1} which are very distinct and easily identifiable. The peak at 1496 cm^{-1} does not show a difference between the trans- and cis-state as

Our response:

We thank the reviewer for raising this point. However, we respectfully disagree - in our opinion the photoswitching related intensity changes in the ATR-FTIR spectrum can be clearly observed in **Figure S1** as shown below. One can clearly distinguish the mentioned peaks at 1602 and 1580 cm^{-1} related to the *trans*-state of the breathing mode with increased intensity compared to the *cis*-state. Moreover, the *cis*-related peak at 1511 cm^{-1} can be easily identified as shoulder. The peak at 1496 cm^{-1} does not

show a difference between the *trans*- and *cis*-state. However, the same behavior for the 1496 cm^{-1} resonance is seen in the above-mentioned publication (Crea et al., *Front. Mol. Biosci.*, 9, 905306, 2022)

We further want to highlight the fact that the recorded nano-FTIR spectra is chemically specific. The spectra have been recorded on top of a lipid vesicle composed of a 50:50% mixture of DOPC and azo-PC with a spectral resolution of 14.2 cm^{-1} . Thus, the spectra are a convolution of the DOPC and the azo-PC spectrum. In addition, during the photoswitching process, not all the azo-PC molecules can be switched in the *cis*-state. Still the nano-FTIR spectrum in **Figure 1f** (reproduced below for clarity) shows a clear difference in intensity at the most prominent photoswitching peaks located at 1606 cm^{-1} , 1463 cm^{-1} and 1495 cm^{-1} . Moreover, both spectra exhibit the characteristics carbonyl resonance located at 1743 cm^{-1} and 1463 cm^{-1} , confirming the chemical nature of a lipid. Therefore, it is in our opinion valid to state that the method allows chemical recognition of a sample.

Actions taken:

We have labeled the specific photoswitching peaks of the ATR-FTIR spectrum shown in **Figure S1** to make it more ease to discern the respective resonances affected by photoswitching. In addition, we modified the caption of **Figure S1** to underline that the respective spectra are recorded on pure DOPC and pure azo-PC.

Reviewer comment:

2. Applicability to real systems. The system and the chemical environment chosen here will have poor applicability in the real systems because of (1) using D₂O instead of H₂O or a buffer because buffer can alter the photoswitching process, however, I agree that H₂O hinders the ring breathing modes in the spectra, (2) the 50% ratio of the azo-lipid seems too high. Authors should explain why they have considered 50% of azo-PC in their study instead of a smaller ratio which could have been more realistic. To my understanding at 50:50 ratio the vesicle could be unstable and vulnerable to any foreign molecule it might encounter during the drug delivery process and hence loses its applicability. Measurement of the efficiency of drug release upon switching could have demonstrated the potential application of this system.

Our response:

We thank the reviewer for being interested in the applicability of the method in real systems. First, we want to emphasize that the method works in principle with any liquid as long as the SiN membrane is stable. Furthermore, we want to bring up the nano-FTIR spectrum of the lipid vesicle recorded in normal H₂O shown in **Fig. S2 b**. This spectrum demonstrates that lipid vesicles can be investigated in normal H₂O, or a buffer solution made from standard H₂O. As the reviewer rightly states we only employed D₂O as the most pronounced photoswitching resonance is located at 1606 cm⁻¹.

We further thank the reviewer for pointing out that the choice of a 50:50 azo-PC/DOPC ratio was not properly addressed in our previous submission. In general, azo-PC is by itself a membrane forming lipid. Liposomes and even supported bilayer membranes can be prepared entirely from azo-PC, but azo-PC doping of regular lipid vesicles made of DOPC, DPhPC or POPC is also possible. In fact, we have studied the properties of vesicles formed with various DOPC/azo-PC compositions in the past. By micropipette aspiration measurements, we found that the change in bending modulus due to photoisomerization is the largest for a DOPC vesicle doped with 50% azo-PC (Urban et al. Langmuir 2018, 34, 13368 – 13374, supporting information Figure S6). Since we aim to observe both spectral changes and mechanical deformation of the liposomes because of the isomerization process, a 50:50 DOPC/azo-PC mixture provides the clearest results. This was not properly explained, so far, but we have now included a paragraph explaining this experimental detail.

Actions taken:

We added the following sentence on **page 4** to the result part of the manuscript to explain our choice of azo-PC:DOPC content for the vesicle formulation:

"In general, azo-PC is a membrane forming lipid and both stable liposomes and supported bilayer membranes can be formed entirely from pure azo-PC. However, the 50:50 % lipid mixture was chosen,

since highest change in bending rigidity is expected based on previous micropipette aspiration measurements²⁸. That way, both spectral changes due to photolipid isomerization, as well as mechanical vesicle deformations are expected, which aids in showcasing the full characterization potential of our approach."

Reviewer comment:

3. It is disappointing to see that the materials and method section does not properly describe the laser parameters, e.g , whether it is a continuous wave laser or a pulsed laser, the pulse length and frequency if it is a pulsed laser. Also, the manuscript fails to describe the underlying theory of the sSNOM method which is the primary focus of the entire manuscript. If it is a pulsed laser, the usual practice is to set the frequency of the laser pulse to be more than four or six times the frequency of the AFM tip depending on whether the 2nd order or 3rd order harmonic demodulations have been considered (see Nat. Commun. 7, 13212, 2016 by Wang et al.). Authors need to carefully address this issue.

Our response:

We agree with the review that we did not describe the laser parameters in enough detail. In the case of the Stuttgart instruments laser employed for conducting the imaging and time traces the laser is pulsed with a 40 MHz frequency and a pulse duration of 400 fs. The nano-FTIR laser employed for near-field spectroscopy is pulsed with 40 MHz repetition rate and a pulse duration of 100 fs. As the reviewer rightly states the pulse frequency of the lasers need to be higher than the tapping frequency of the s-SNOM. For the second-harmonic demodulation with a lock-in amplifier, the sampling requires the pulse repetition rate to be four times higher than the tip oscillation frequency. This is easily fulfilled by the lasers of the optical setup, and they can be regarded as quasi-continuous wave lasers. We further want to highlight that the publication cited by the reviewer is a very special case of s-SNOM operation based on low-repetition-rate pulsed light sources. As already stated in the abstract of the publication s-SNOM is normally operated with high repetition-rate pulsed lasers as utilized in this study.

Explaining the s-SNOM method in even more detail than given in the method section of the manuscript is in our opinion beyond the scope of this publication as it is a research paper that is focused on introducing the new concept of transient infrared nanoscopy in liquid. We refer the reviewer to

several cited papers including two reviews on s-SNOM for the exact description of the technical and theoretical concepts of s-SNOM:

Review:

Keilmann, F. & Hillenbrand, R. Near-field microscopy by elastic light scattering from a tip. *Philosophical transactions. Series A, Mathematical, physical, and engineering sciences* 362, 787–805; 10.1098/rsta.2003.1347 (2004).

Chen, X. et al. Modern Scattering-Type Scanning Near-Field Optical Microscopy for Advanced Material Research. *Advanced materials (Deerfield Beach, Fla.)* 31, e1804774; 10.1002/adma.201804774 (2019).

Methodological publications:

Amarie, S. & Keilmann, F. Broadband-infrared assessment of phonon resonance in scattering-type near-field microscopy. *Phys. Rev. B* 83; 10.1103/PhysRevB.83.045404 (2011).

Hillenbrand, R., Knoll, B. & Keilmann, F. Pure optical contrast in scattering-type scanning near-field microscopy. *Journal of microscopy* 202, 77–83; 10.1046/j.1365-2818.2001.00794.x (2001).

Vicentini, E. et al. Pseudoheterodyne interferometry for multicolor near-field imaging. *Optics express* 31, 22308–22322; 10.1364/OE.492213 (2023).

Huth, F. et al. Nano-FTIR absorption spectroscopy of molecular fingerprints at 20 nm spatial resolution. *Nano letters* 12, 3973–3978; 10.1021/nl301159v (2012).

Amarie, S., Ganz, T. & Keilmann, F. Mid-infrared near-field spectroscopy. *Optics express* 17, 21794–21801; 10.1364/OE.17.021794 (2009).

Hauer, B., Engelhardt, A. P. & Taubner, T. Quasi-analytical model for scattering infrared near-field microscopy on layered systems. *Optics express* 20, 13173–13188; 10.1364/OE.20.013173 (2012).

Actions taken:

We added the following sentences to the Materials and Methods section of the manuscript on **page 18** and to clarify the laser parameters for the reader of the manuscript:

"The Stuttgart Instrument laser works with a pulse rate of 40 MHz and a temporal pulse duration of 400 fs."

"The nano-FTIR laser operates at 80 MHz repetition rate delivering 400 fs pulses."

"A detailed description of the fundamental working principle of s-SNOM is, e.g., provided in the following literature reviews^{12,13}."

Reviewer comment:

4. Also, the thermodynamic characterizations (such as phase transition) of this vesicle has not been done, so the phase of the system is completely unknown. Since the switching process highly depends on the lipid phase it is crucial to characterize the thermodynamics of the vesicle prior to the photoswitching study. There are already some nice literatures available in the similar direction. Such as by Quaroni et al. *RSC Advances* 8, 2786 (2018), and by Cernecu et al., *Analytical Chemistry* 90, 10179

(2018), and by Roman et al., *Nanotechnology* 30, 425502 (2019). They nicely described the spectroscopy and/or imaging data from lipid droplets, vesicles and also from the single cell. Authors should describe the speciality of their method used in this study and also, if possible, should complement their results with some other methods (e.g., Small angle X-ray scattering).

Our response:

We thank the reviewer for this question. Azo-PC does not undergo a phase transition at temperatures where our experiments were conducted and both *trans*- and *cis*-isomers form a fluid bilayer membrane.

The azo-PC membrane system has been extensively characterized in the past, with different methods including also small angle x-ray scatterings (SAXS) and NMR, as described in the following publications

[SAXS] M. F. Ober, A. Müller-Deku, A. Baptist, B. Ajanović, H. Amenitsch, O. Thorn-Seshold, B. Nickel, SAXS measurements of azobenzene lipid vesicles reveal buffer-dependent photoswitching and quantitative Z→E isomerisation by X-rays. *Nanophotonics* 11, 2361–2368 (2022).

[NMR] *J Am Chem Soc* . 2021 Jun 16. doi: 10.1021/jacs.1c03524. : “*How Photoswitchable Lipids Affect the Order and Dynamics of Lipid Bilayers and Embedded Proteins*” Mahmoudreza Doroudgar 1, Johannes Morstein 2, Johanna Becker-Baldus 1, Dirk Trauner 2, Clemens Glaubitz 1Pernpeintner, C. et al. Light-Controlled Membrane Mechanics and Shape Transitions of Photoswitchable Lipid Vesicles.

Langmuir : the ACS journal of surfaces and colloids 33, 4083–4089; 10.1021/acs.langmuir.7b01020 (2017).

Pritzl, S. D. et al. Photolipid Bilayer Permeability is Controlled by Transient Pore Formation. Langmuir : the ACS journal of surfaces and colloids 36, 13509–13515; 10.1021/acs.langmuir.0c02229 (2020).

Urban, P. et al. Light-Controlled Lipid Interaction and Membrane Organization in Photolipid Bilayer Vesicles. Langmuir : the ACS journal of surfaces and colloids 34, 13368–13374; 10.1021/acs.langmuir.8b03241 (2018).

Osella, S., Granucci, G., Persico, M. & Knippenberg, S. Dual photoisomerization mechanism of azobenzene embedded in a lipid membrane. Journal of materials chemistry. B 11, 2518–2529; 10.1039/D2TB02767D (2023).

Kuiper, J. M. & Engberts, J. B. F. N. H-aggregation of azobenzene-substituted amphiphiles in vesicular membranes. Langmuir : the ACS journal of surfaces and colloids 20, 1152–1160; 10.1021/la0358724 (2004).

Kuiper, J. M., Stuart, M. C. A. & Engberts, J. B. F. N. Photochemically induced disturbance of the alkyl chain packing in vesicular membranes. Langmuir : the ACS journal of surfaces and colloids 24, 426–432; 10.1021/la702892m (2008).

Valley, D. T., Onstott, M., Malyk, S. & Benderskii, A. V. Steric hindrance of photoswitching in self-assembled monolayers of azobenzene and alkane thiols. Langmuir : the ACS journal of surfaces and colloids 29, 11623–11631; 10.1021/la402144g (2013).

Zhang, J. et al. Label-Free Time-Resolved Monitoring of Photolipid Bilayer Isomerization by Plasmonic Sensing. Advanced Optical Materials 12; 10.1002/adom.202302266 (2024).

In addition, the publications listed by the reviewer show measurements of dried and fixed lipid samples without any dynamic. We even cited Cernecu et al., Analytical Chemistry 90, 10179 (2018) as an example of the unique advantages of s-SNOM in the field of biospectroscopy. However, our manuscript goes significantly beyond the listed publications as we conduct our experiments in liquid. Moreover, we demonstrate a method to characterize dynamic lipid processes in liquid on the nanoscale, which has never been done before with near-field microscopy. In our opinion, the following paragraph in the introduction explains the points of novelty:

“Here, we demonstrate the use of in-situ nanoscopy to image and spectroscopically analyze individual photoswitchable lipid vesicles with sizes down to 176 nm in aqueous environments. In contrast to previous investigations on naturally progressing biological systems, we present the first in-situ s-SNOM study on actively induced dynamic processes by reversibly changing the morphology of a vesicle through repeated ultraviolet/blue light illumination and tracking its spectral response at 50 ms temporal resolution. Our method demonstrates not only the possibility to detect and distinguish two photoisomeric states of the lipid molecules on the single lipid vesicle level based on subtle changes in

their near-field MIR spectra, but also monitor the photoinduced transformations of the lipids in their aqueous environment in real time.”

Actions taken:

We want to further clarify the novelty of our manuscript and added the following sentence in the discussion section of our manuscript on **page 15**:

"This goes significantly beyond current state-of-the-art near-field microscopy studies on lipids, since the spectroscopy, imaging and transient signal tracing were conducted in liquid and even dynamic processes have been recorded."

Reviewer comment:

5. Page 3. Authors described the advantages of using SiN membrane. But the possible disadvantages, such as the potential hindrance of the near-field interactions by SiN, have not been discussed.

Our response:

We agree with the reviewer that the liquid method needs to be set in context of the whole scientific literature of IR s-SNOM in liquid and better discuss the advantages and disadvantages of different liquid s-SNOM approaches.

Actions taken:

We added the following paragraph to the discussion section of the manuscript on **page 16** to also acknowledge the inherent disadvantage of the method:

"There are important performance differences between our membrane based liquid s-SNOM method and competing methods, which include in-liquid s-SNOM. In general, approaches that measure s-SNOM or AFM-IR directly in liquid can provide spatial resolutions only limited by the tip and recording of the full topography⁵⁰⁻⁵⁴, whereas membrane-based method is limited to around 100 nm. However, these approaches all employ an ATR-based transmission illumination, which is very difficult to align. Moreover, our method enables more robust and longer duration experiments, because the tip is protected from contamination by the sample and the optical alignment is maintained, allowing more complex dynamic studies of, e.g., living cells¹⁸. Thinner membranes could be utilized to further increase the spatial resolution limit of our method, as long as they provide sufficient mechanical stability for water adhesion and AFM tapping. For example, liquid s-SNOM studies have been conducted with ultra-thin graphene capping layers⁵⁵⁻⁵⁷ and metal oxide capping layers such as TiO₂ and Al₂O₃⁵⁸, which enable higher resolution imaging in liquid. Building on this idea, it has been demonstrated that a liquid sample containing viruses can be wrapped in a graphene sheet and investigated with s-SNOM enabling true nanoscale imaging and topography recording with a capping layer⁵⁹. These capping layers have the additional benefit of having no phonon resonance in the MIR compared to SiN, which exhibits a strong resonance in the range of 800 to 1150 cm⁻¹ masking important molecular infrared resonances¹⁸. From a technological perspective, the wide and cheap commercial availability of the SiN membranes enables straightforward consecutive experiments on

complex biological systems, as opposed to other capping layers, which need to be self-fabricated for every experiment."

Reviewer comment:

6. Page 10, Figure 3. Whether any sort of corrections have been applied on the obtained images for enhancing the contrast should be described. Also, according to my understanding, the synchronization of detector sampling rate with the pulse rate of the laser source plays a great role to maintain the near-field contrast. Did the author consider these points?

Our response:

We thank the review for being interested in our data processing techniques. We did not use any data processing for the **Figure 3a** and **b**. They display the recorded images attained as data files from the commercial NeaScope operated in the Ps-Het near-field imaging mode. **Figure 3c** shows the data processing from the data displayed in **Figure 3a** used for the extraction of the circularity value. Here, we applied a threshold of $s_2 = 7.8$ a.u. for the recorded values to define the boundary between the liquid and the vesicle. The value was chosen as the optical amplitude is above $\frac{1}{e}$ of its maximum value at the particle center (after background subtraction). Concerning the detector sampling rate and the relation to the pulse frequency of the laser. We want to clarify how the internal data processing of the s-SNOM device work. The detector signal is sampled at a high enough bandwidth from the data acquisition card (DAQ) of the microscope to enable a 4th order demodulation of the recorded signal at harmonic ($n\Omega$) of the tip frequency Ω . The pulse rate of the laser employed for imaging is around 40 MHz. Thus, the laser pulse rate is much higher than the sampling rate of the detector and the recorded near-field contrast does not depend on a careful tuning of the laser pulse rate to the detector sampling rate. The main mechanism to attain the near-field imaging contrast is demodulation of the signal at higher harmonics of the tip and the Ps-Het processing of the tip signal. Moreover, we want to emphasize that this is a standard measurement mode of a commercial system that is well established in the scientific community. As mentioned above there are several papers and reviews explaining the concepts and theory behind the microscope and it is not the aim of the paper to explain these details.

Actions taken:

We added the following sentence to the materials and method section of the manuscript on **page 18** to clarify the sampling rate of the detector and the relationship to the demodulation procedure of the internal lock-in amplifier:

"Subsequently, the backscattered light is detected by a nitrogen-cooled mercury cadmium telluride (MCT) detector (IR-20-00103, *Infrared Associates Inc.*, Stuart, USA), which is sampled by the internal data acquisition card (DAQ) at a high enough bandwidth to demodulate the recorded signal at harmonics $n\Omega$ of the tapping frequency Ω by the DAQ electronics in order to separate the near-field signal from unwanted far-field signals."

In addition, we added the following sentence to the main part of the manuscript on **page 9** to explain the threshold value for **Figure 3c**:

"This value was chosen to encompass the area where the optical amplitude is above $\frac{1}{e}$ of its maximum value at the particle center after background subtraction."

Reviewer comment:

7. Page 11. The statement “switching dynamics at a modest SNR of about 4 down to around 50 ms (Figs. S8, S9, S10)”. Figs. S8, S9, S10 suggest that the switching dynamics are a few tens of seconds long whereas it is claimed that the time resolution is less than 50 ms. Do they mean temporal resolution as the step size used in the measurement? Do the authors have any reference measurements of the temporal resolution? Also, since the entire result and the subsequent claims are primarily based on the switching dynamics, I suggest to fit the data shown in Figs. S8 and S9 with a transient equation, and to move them to the main manuscript with the discussions of the findings.

Our response:

We thank the reviewer for the comment on the transient time trace section of the result part. The reviewer is correct in his assessment that the switching dynamics is in the range of a few tens of seconds as can be seen in **Fig. 4, S5, S6, S11, S12**. We agree that a fit of the data is important and we choose to fit the transient time trace shown in **Fig. 4** as it is the time trace with the highest quality and shows four reproducible switching cycle in one measurement, which enables us to extract a time constant τ and a delay time t_d with an associated error, which is not possible for the measurements in the SI section. Another reviewer asked us to fit this time trace and we think it is a good idea as gives interesting information relating to the switching dynamics.

The time constant of t_p is based on an average of the continuously read-out of the Ps-Het signal over the temporal interval of t_p . This means that each data point represents an average value over t_p of the signal trace signal, and consecutive data points are spaced by t_p . The values are based on the software of the lock-in amplifier of the Ps-Het detection mode of the commercial s-SNOM microscope and should not need a reference measurement as they are guaranteed by the manufacturer. As the switching dynamics of the lipid vesicle is in the order of several seconds the recorded switching dynamics is not convoluted as it is much slower.

We do not want to include the high-resolution switching dynamics measurements in the main part of the manuscript as we believe **Fig. 4** is clearer to understand, when we solely show the time trace that tracks the continuous back-and-forth switching over four full cycles.

Actions taken:

We added a sinusoidal fit (red curve) to the time traces shown in **Fig. 4**:

Figure 4: Resolving the photoswitching dynamics of a single 500 nm nanoscale lipid vesicle by millisecond-MIR near-field signal traces. (a) Optical near-field amplitude image normalized to the surrounding D₂O ($S_2/S_{2,D_2O}$) and (b) near-field phase image (φ_2), recorded at 1603 cm⁻¹ with 2 min acquisition time, directly before starting the MIR signal trace acquisition at the position of the cross, scale bars 500 nm. The recorded amplitude ($S_2/S_{2,D_2O}$) in (c) together with the phase (φ_2) in (d) reveal the photoswitching dynamics of the eight switching events at a temporal resolution of $t_p = 500$ ms. The symbols above (c) in combination with the dashed vertical lines indicate when the illumination wavelength was switched to the value written above. The blue and violet coloring of the signal traces mark the switching light's wavelength at each point. The black curves are moving averages of the measured points, and the red curves are sigmoidal fits of the switching process of the form $f(t) = \frac{L}{1 + e^{-\frac{(t-t_d)}{\tau}}} + C$ as displayed in detail in Fig. S5 and S6. (e) Optical amplitude and (f) phase images taken directly after acquiring the near-field traces show that the vesicle remained stable in both position and signal strength, even after eight consecutive switching transitions, scale bars 500 nm.

Furthermore, we added two more Figures to the supporting information displaying the fit of each switching event in details for the amplitude trace (Fig. S5) and the phase trace (Fig. S6) with the accompanying growth parameters τ and delay time t_d

Figure S5: Sigmoidal fit of the transient amplitude signal time trace for investigating the photoswitching dynamic shown in Fig. 4. The extracted time constants τ determines the steepness of the switching behaviour describing how fast the system responds. The average τ -value for the *trans-to-cis* switching is 2.8 ± 1.1 s, whereas the τ -value for the *cis-to-trans* is 1.5 ± 0.5 s, indicating that the *cis-to-trans* switching process occurs faster. The delay time t_d , defined as the interval between the switching of the light and the inflection point of the fitted curve, is 11.8 ± 1.0 s for *trans-to-cis* switching and 13.0 ± 2.6 s for *cis-to-trans* switching. The fitting was performed with a sigmoidal function of the following form $f(t) = \frac{L}{1 + e^{-\frac{t-t_d}{\tau}}} + C$.

Figure S6: Sigmoidal fit of the transient phase signal time trace for investigating the photoswitching dynamic shown in Fig. 4. The extracted growth parameters τ determines the steepness of the switching behaviour describing how fast the system responds. The average τ -value for the *trans*-to-*cis* switching is 3.0 ± 0.6 s, whereas the τ -value for the *cis*-to-*trans* switching is 1.6 ± 0.8 s, further indicating that the *cis*-to-*trans*-switching occurs faster. The delay time t_d is 11.7 ± 1.9 s for *trans*-to-*cis* switching and 13.6 ± 2.7 s for *cis*-to-*trans* switching. The fitting was performed with the same sigmoidal function as in Figure S5.

We added the following sentences to the result part of the manuscript on **page 12** and **13** to explain the findings of the fits:

"When fitting the MIR signals of the switching events in **Figs. 4c, d** with a sigmoidal response function (**Figs. S5 and S6 and red curves in Fig. 4c, d**), characteristic delay times t_d ranging from 10.3 to 15.7 s and growth parameters τ ranging from 0.7 to 3.9s could be extracted."

"The extracted growth parameters τ through a sigmoidal fit of the time traces (**Fig. S5, S6**) indicate a higher switching speed in the *cis-to-trans* direction in comparison to the *trans-to-cis*-direction, in agreement with previous reports. ~~In the measurements presented in Fig. 4, however, it seems that both isomerization rates are almost similar.~~"

We added the following sentence to the method section of the manuscript on **page 18** to better explain the time resolution of the transient signal traces:

"The integration time t_p of the Ps-Het based transient signal trace method defines the time interval at which the continuously read-out Ps-Het data points are averaged and the spacing of consecutive data points in time."

Response letter to reviewer comments

Reviewer #1 (Remarks to the Author):

General statement:

The authors have sufficiently addressed many of my concerns.

The authors have not yet addressed my concerns about the scientific interpretation of their time-resolved measurements. The authors use a sigmoid equation lacking citation. The authors claim without evidence or sufficient citation that the appearance of such a sigmoid in time-resolved data justifies a claim of cooperative photoswitching.

The authors have also not yet addressed my concerns about their experimental time resolution.

Pending these minor corrections, this manuscript will be suitable for publication in Nature Communications.

Detailed below related to my original review items 1 and 6:

Our response:

We sincerely appreciate the time and effort the reviewer dedicated to evaluating our work and for providing constructive suggestions on how to improve the quality of our manuscript. We are pleased to report that we have carefully considered all of the questions and suggestions raised by the reviewer and have revised our manuscript accordingly.

Reviewer comment:

1:

Thank you for the clarification. I believe the scan length reported in the author response (350 μm) is correct, but the scan length reported in the edited manuscript (300 μm) is a typographical error. If the scan length is 350 μm , then I believe the spectral resolution is $1/0.07=14.2857\text{ cm}^{-1}$, which rounds to 14.3 cm^{-1} not 14.2 cm^{-1} as written.

Our response:

We thank the reviewer for pointing out this mistake. The scan length is indeed 350 μm and our spectral resolution is therefore 14.3 cm^{-1} . This is now corrected in the revised manuscript.

Actions taken:

We corrected the following passage in the Materials and Methods section:

“The nano-FTIR spectra shown in Fig. 1 were recorded with an interferometer scan length of 300 350 μm , resulting in a nominal spectral resolution of 14.2 14.3 cm^{-1} ”

Reviewer comment:

6:

I believe in their response, the authors erroneously reported the illumination as 0.48 mW m⁻². I believe the value should be 0.48 MW m⁻². I encourage the authors to confirm the correct value and include this information in the revised manuscript or supplement.

I believe the word sinusoidal in the author response was a typographical error and the authors intended to write sigmoidal.

The authors state in their reply "The time constant of 500 ms is based on a 500 ms average of the continuously read-out Psu-Het signal" and then claim "the recording switching dynamics is not convoluted as it is much slower." Let me be clear: observed dynamics are necessarily convoluted with instrument response time in ANY experiment. I would argue strongly that an instrumental time resolution of 0.5 seconds will absolutely affect the value of tau, which is reported in the manuscript as "ranging from 0.7 to 3.9s." Furthermore, depending on data processing, a 500 ms running average could result in an instrument resolution closer to 1 s, depending on implementation of the moving average. The authors must absolutely address experimental uncertainty before making claims based upon their fits.

The authors state in the original manuscript and in their reply that a sigmoidal is indicative of a cooperative response. The authors need to provide a citation clearly justifying the connection between a sigmoid and a cooperative response. I have no doubt that the photoswitching is related to a first order phase transition, but I do not see the connection specifically to the mathematical function of a sigmoid. (If the switching is associated with a first order phase transition, then I naively would expect a variable induction time, which the authors fit include in their sigmoid as term t_d, but I do not see an obvious connection to the shape of the sigmoid and the fitted tau parameter.) The authors must include a citation clearly justifying the use of the sigmoid function and explain how the appearance of a sigmoidal justifies claims of cooperative behavior before drawing scientific conclusions from their fit parameters.

Our response:

We thank the Reviewer for highlighting these important concerns, which were not sufficiently addressed in the previous version of our manuscript and response letter. In line with the Reviewer's suggestions, we have now clarified the ambiguities related to experimental uncertainties and added a citation to support the use of a sigmoidal fit function as justification for cooperative behavior among photolipids. Specifically, the following revisions have been made:

Illumination intensity

We thank the Reviewer for pointing out the error in our estimation of the illumination intensity. As stated in the *Materials and Methods* section of the manuscript, the estimated illuminated area is approximately 500 μm × 500 μm. Based on this area, the corrected irradiance is calculated as follows:

$$E = \frac{P}{A} = \frac{30 \text{ mW}}{500 \mu\text{m} \times 500 \mu\text{m}} = 12000 \frac{\text{mW}}{\text{cm}^2}$$

Deconvolution of observed dynamics and instrument response time

The reviewer is correct in pointing out that, while the time between both cis and trans steady state is much longer than the sampling rate used for our experiments, the value of τ extracted from the sigmoidal fits is much closer, being even below 1s in some measurements. Because the sampling rate and τ are on the same order of magnitude, we agree with the reviewer that the convolution with the instrument response has to be taken into account. It likely means that our fitted switching times overestimate the true photoswitching dynamics.

However, we would argue that, since we have obtained values of τ that are almost eight times higher than the sampling time, the observed photoswitching response is not merely the result of a convoluted exponential decay. Rather, it can genuinely be well-described by a sigmoidal function, reflecting the inherent characteristics of the underlying switching process.

Running average

Regarding the running average plotted together with the fits: the sigmoidal function was fitted to the data points directly and not to the running average. Therefore, the running average does not affect the obtained parameters.

Cooperative switching behavior and sigmoidal functions

To the best of our knowledge, the specific system we studied in our work (photoswitching AZO-based photolipids) has not been described using sigmoidal functions in the literature before. However, there are several publications that have demonstrated cooperative processes described by sigmoidal functions in related systems. One study analyzed the time dynamics of cooperative binding between giant unilamellar vesicle (GUV) membranes and polymers and directly linked the appearance of a sigmoidal response to cooperative effects (<https://doi.org/10.1007/s00232-022-00220-6>).

Similarly, another study concluded that the mechanism behind polymer-induced bursting of GUVs can also be described by a sigmoidal function (<https://doi.org/10.1021/acs.langmuir.1c01047>). Here the authors stated that this reaction occurred due to “the accumulation of the polymer chains with membrane-active conformations with time”, resulting in a positive feedback loop driven by polymer-lipid bilayer interactions.

As we have mentioned in the manuscript, cooperative behavior has been well established in azo-PC lipid bilayer membranes. It is thus reasonable to interpret the sigmoidal shapes of our measurement curves as indicative of such processes. While more complex modeling, ideally accompanied by molecular dynamics simulations, may be appropriate to fully capture the temporal dynamics of the system, this lies beyond the scope of this work. However, this represents a promising direction for future research.

Actions taken:

We have added the calculated value of the irradiance to the Materials and Methods section:

“This yields an irradiance of $E = 12000 \frac{\text{mW}}{\text{cm}^2}$.”

Furthermore, we addressed that the resulting fits are a convolution between instrument response time and the actual photoswitching process and how this will affect the results:

“Note that the extracted photoswitching times are inherently convoluted with the sampling time of 0.5 s. Given that the observed growth parameters are approximately on the same order (between 0.7 s to 3.9 s) as the instrumental resolution, we expect the effect of convolution on the fitted values for τ to be non-negligible. This means that the actual switching dynamics are likely faster than the fits would suggest, especially for lower values of τ that are closer to the instrumental response time.”

To strengthen our claim that a sigmoidal response shows evidence of cooperative switching behavior, we added the following section below Fig. 4:

“The appearance of sigmoidal response functions has been previously demonstrated in related systems to describe cooperative effects, such as e.g. binding between giant unilamellar vesicle (GUV)

membranes and polymer through domain formation [<https://doi.org/10.1007/s00232-022-00220-6>], which has directly been linked to the presence of cooperative processes. Similarly, the dynamics of polymer-induced bursts of GUVs have also been found to behave sigmoidally due to the accumulation of polymer chains in the vicinity of the bilayer surface leading to a cooperative behavior that causes a sudden rupture in bilayer membranes [<https://doi.org/10.1021/acs.langmuir.1c01047>].”

The following new references are now included in our revised MS:

<https://doi.org/10.1007/s00232-022-00220-6>

<https://doi.org/10.1021/acs.langmuir.1c01047>

Reviewer #3 (Remarks to the Author):

General statement:

The authors have addressed my comments in detail. I can recommend this manuscript to be published on Nature Communications, pending the following minor comments for the benefit of readers:

Reviewer comment:

1. 30 ms temporal resolution --- as Reviewer #4 pointed out, this could be misleading, as readers may incorrectly interpret it as indicating actual signal changes, which in reality happen between 1 to 2 seconds.

To clarify, perhaps on p.13 or p.11, the authors could add a simple sentence explicitly stating that this refers to the time per sampling step required to obtain the time-domain interferogram in nano-FTIR measurements. Also, note that p.19, line 544 states "35 ms per point".

Our response:

We thank the Reviewer for making us aware that a reader may potentially confuse temporal resolution and the time constants associated with changes in signal due to photoswitching. Regarding line 544 on page 19, the sentence refers to the temporal resolution of the nano-FTIR spectra acquisition and not to the single-wavelength tracking of the photoswitching dynamics. Thus, this value refers to the data presented in Fig. 1.

Actions taken:

To avoid confusion between the sampling steps of the transient nanoscopy method and the temporal resolution of the nano-FTIR spectral acquisition, we have replaced the term “temporal resolution” with “sampling time” in multiple instances throughout the manuscript, as well as in the figures:

“Our transient nanoscopy approach enables imaging below the diffraction limit and tracks dynamics with ~~temporal resolution~~ sampling times as fast as 30 ms.”

“In contrast to previous investigations on naturally progressing biological systems¹⁸, we present the first in-situ s-SNOM study on actively induced dynamic processes by reversibly changing the morphology of a vesicle through repeated ultraviolet/blue light illumination and tracking its spectral response at 30 ms ~~temporal resolution~~ sampling time.”

“This approach allows to resolve the switching dynamics at a ~~temporal resolution~~ sampling time down to $t_p = 30$ ms with SNR of 4 (Figs. S10, S11, S12).”

“The s-SNOM signals are continuously recorded at 1603 cm^{-1} with $t_p = 500\text{ ms}$ **sampling time** (Fig. 4c, d).”

“The recorded amplitude ($S_2/S_{2,D2O}$) in (c) together with the phase (φ_2) in (d) reveal the photoswitching dynamics of the eight switching events at a **temporal resolution-sampling time** of $t_p = 500\text{ ms}$.”

“The signal-to-noise characteristics allowed to detect a photoswitching signature at a **temporal resolution sampling time** as short as $t_p = 30\text{ ms}$ with a statistical significance limit of 3.72σ (see Fig. S12 and Figs. S10 and S11 for exact values).”

Additionally, we clarified in the Materials and Methods section that the 35 ms value refers to the spectra shown in Fig. 1:

“The nano-FTIR spectra **shown in Fig. 1** were recorded with an interferometer scan length of **300 350** μm , resulting in a nominal spectral resolution of **14.2 14.3** cm^{-1} .”

Reviewer comment:

2. As the authors pointed out in their response letter, finite dipole model is less useful for complex unknown biological samples. It will be interesting to discuss alternative approaches to mitigate this. I would like to refer the authors to a relevant overlooked review paper on THz nanoscopy (Appl. Phys. Rev. 11, 021306 (2024) <https://doi.org/10.1063/5.0189061>), which proposes an envisioned measurement procedure (Fig. 12 in that reference) in light of Sci. Rep. 11, 21860 (2021).

Our response:

We agree that our work would benefit from mentioning alternative methods to model more complex biological samples. The work proposed by the reviewer provides an excellent iterative approach for achieving this task.

Actions taken:

We added the following passage in the Discussion section on page 16:

“Additionally, more sophisticated approaches may be used in future studies to extract the permittivity and model the optical response of various biological materials adhering to the membrane through iteratively measuring and optimizing the liquid cell [<https://doi.org/10.1063/5.0189061>], which can serve as a tool to analytically calculate and predict the response of complex samples with an unknown dielectric function.”

The following new reference is now included in our revised MS:

<https://doi.org/10.1063/5.0189061>

Reviewer comment:

3. Since s-SNOM probes local dielectric constants, it would be helpful for non-chemistry readers if the authors could comment on how relative changes in observed local dielectric constants may be interpreted in terms of biochemical processes in liquid environments.

Our response:

We thank the reviewer for this helpful suggestion and agree that a connection between alterations in bond structure and dielectric function should be provided.

Actions taken:

We added the following paragraph above Fig. 4 on page 11:

“Both infrared signals originate from repeated changes in the local dielectric function of the material underneath the tip. [<https://doi.org/10.1063/1.4905507>] The dielectric function in the present case of organic soft matter is determined by intra- as well as inter-molecular bonds [https://doi.org/10.1007/978-3-642-93186-4_1] of the phospholipids. Quite generally, biochemical processes in liquid environment can be tracked quantitatively by our transient nanoscopy method, provided they change the dielectric function at a selected MIR frequency sufficiently strongly to overcome the instrument noise.”

The following new references are now included in our revised MS:

https://doi.org/10.1007/978-3-642-93186-4_1

Response letter to reviewer comments

Reviewer #1 (Remarks to the Author):

General statement:

The authors have addressed my comments. I recommend this manuscript for publication in Nature Communications.

Our response:

We thank the reviewer for considering that our manuscript is of high enough quality to recommend it for publication in Nature Communication.